# Charged peptides enriched in aromatic residues decelerate condensate ageing driven by cross-$\beta$-sheet formation

Ignacio Sanchez-Burgos [1,2], Andres R. Tejedor [2,3], Alejandro Castro[3], Alejandro Feito [3], Rosana Collepardo-Guevara [1,2,4] ✉ & Jorge R. Espinosa [1,3] ✉

Biomolecular condensates play wide-ranging roles in cellular compartmentalization and biological processes. However, their transition from a functional liquid-like phase into a solid-like state—usually termed as condensate ageing—represents a hallmark associated with the onset of multiple neurodegenerative diseases. In this study, we design a computational pipeline to explore potential candidates, in the form of small peptides, to regulate ageing kinetics in biomolecular condensates. By combining equilibrium and non-equilibrium simulations of a sequence-dependent residue-resolution force field, we investigate the impact of peptide insertion—with different composition, patterning, and net charge—in the condensate phase diagram and ageing kinetics of archetypal proteins driving condensate ageing: TDP-43 and FUS. We reveal that small peptides composed of a specific balance of aromatic and charged residues can substantially decelerate ageing over an order of magnitude. The mechanism is controlled through condensate density reduction induced by peptide self-repulsive electrostatic interactions that specifically target protein regions prone to form cross-$\beta$-sheet fibrils. Our work proposes an efficient computational framework to rapidly scan the impact of small molecule insertion in condensate ageing and develop novel pathways for controlling phase transitions relevant to disease prevention.

Cell compartmentalization is a cornerstone of cellular organization, enabling spatiotemporal regulation of biochemical reactions with remarkable precision[1]. While membrane-bound organelles have been recognized as the primary means of achieving compartmentalization, transformative experiments have revealed liquid-liquid phase separation (LLPS) as a highly dynamic and versatile mechanism for organizing cellular content[2–4]. Through LLPS, biomolecular condensates are formed via multivalent and dynamic interactions among proteins, nucleic acids, and other macromolecules, creating membraneless phase-separated compartments with unique biochemical properties[1,5–9]. For instance, the formation of the nucleoli—where ribosomal RNA is synthesized and processed—is driven by LLPS of key constituents including fibrillarin and nucleophosmin that facilitate the assembly of ribosomal subunits[10–15]. Moreover, the emergence of stress granules, which appear in response to cellular stress, concentrating mRNAs and proteins[16,17], or the nucleation of P granules, which play essential roles in germ cell development as found in *Caenorhabditis elegans*[18,19], represent fundamental examples of intracellular organization via condensate

[1]Maxwell Centre, Cavendish Laboratory, Department of Physics, University of Cambridge, J J Thomson Avenue, Cambridge CB3 0HE, UK. [2]Yusuf Hamied Department of Chemistry, University of Cambridge, Lensfield Road, Cambridge CB2 1EW, UK. [3]Department of Physical-Chemistry, Universidad Complutense de Madrid, Av. Complutense s/n, 28040 Madrid, Spain. [4]Department of Genetics, University of Cambridge, Cambridge CB2 3EH, UK. ✉ e-mail: rc597@cam.ac.uk; jorgerene@ucm.es

formation. Importantly, biomolecular condensates can rapidly assemble and disassemble, which provides a unique flexible mechanism to regulate essential cellular processes such as gene expression, stress responses, and signal transduction among others[20–22].

Nevertheless, dysregulation of intracellular LLPS has been implicated in the onset of multiple neurodegenerative disorders[23–25] including amyotrophic lateral sclerosis (ALS), Alzheimer's disease, Parkinson's disease, and certain types of dementia−where condensate phase-separation eventually leads to the formation of aberrant solid-like aggregates[26–29]. In ALS, mutations in proteins such as the Transactive response DNA binding protein of 43 kDa (TDP-43) and Fused in Sarcoma (FUS) disrupt their functional phase behaviour, resulting in the formation of toxic solid-like aggregates that compromise cellular function[30–36]. Similarly, in frontotemporal dementia, the misfolding and aggregation of Tau protein and other disease-associated proteins are linked to altered LLPS[37–39], contributing to neuronal dysfunction and cell death. Beyond neurodegenerative diseases, LLPS has also been associated with cancer, where aberrant condensate formation influences oncogenic signalling pathways and promotes tumorigenesis[40–45]. The progressive transition of functional liquid-like biomolecular condensates into pathological solid-like states (e.g., ageing) has been suggested to be originated by the gradual accumulation of inter-protein $\beta$-sheets within the condensates[39,46–48]. These inter-protein structures can be formed over time when biomolecules are concentrated to high levels, as illustrated in Fig. 1a, and are capable of transforming transient liquid-like interactions into long-lived highly energetic cross-links characteristic of kinetically arrested states such as glasses or gels. The progressive accumulation of cross-$\beta$-sheet structures over time has explained the gradual decrease in molecular mobility that numerous biomolecular condensates experience over time[7,49–54]. Therefore, understanding the molecular underpinnings of LLPS in these disorders not only elucidates a fundamental

mechanism of pathogenesis, but also underscores the urgent need to develop strategies to regulate LLPS due to its critical role in health and disease.

To address this challenge, a potential therapeutic strategy has involved the development of small molecules or peptides that specifically modulate LLPS[55–57]. Multiple studies have focused on using different molecules to dissolve already formed condensates[57–61]. Examples include dissolution of DDX3[62], PARP1[63], and PI3K[64] stress granules with the use of synthetic molecules, as well as the dissolution of Tau[65] and p53[66] condensates with the addition of small peptides. Another considerable body of literature focuses on the inhibition or delay in the condensates ageing, rather than dissolving them. Particularly, delaying or inhibiting aggregation in Tau[67], A$\beta$42[68–70], TDP-43[56,71] and $\alpha$-synuclein[72–74] condensates has been attempted in presence of LLPS modulating small organic molecules. Nevertheless, while short peptides offer a promising strategy for therapeutic intervention due to their specificity and potential for cell-penetration[75], the precise protein-peptide interactions required to control the aggregation and ageing pathways remains largely unknown[18,60,76].

Computational approaches represent a powerful tool for expediting the screening and optimization of suitable peptide sequences for this goal, and contribute to the development of effective therapeutics against condensate-related disorders such as neurodegenerative diseases[77]. Here, we perform non-equilibrium Molecular Dynamics (MD) simulations at the coarse-grained residue-resolution level[78] to investigate the inhibition of condensate ageing through the insertion of small peptides at relatively low peptide concentrations. As a proof-of-concept, we focus on the low-complexity domains of FUS (referred to as FUS LCD) and TDP-43 (referred to as TDP-43 LCD) due to their well-established relevance in condensate misregulation and ageing driven by inter-protein $\beta$-sheet accumulation[30–36]. In Fig. 1b, we highlight the charged and aromatic residues within both protein sequences, as well as the low-complexity aromatic-rich kinked

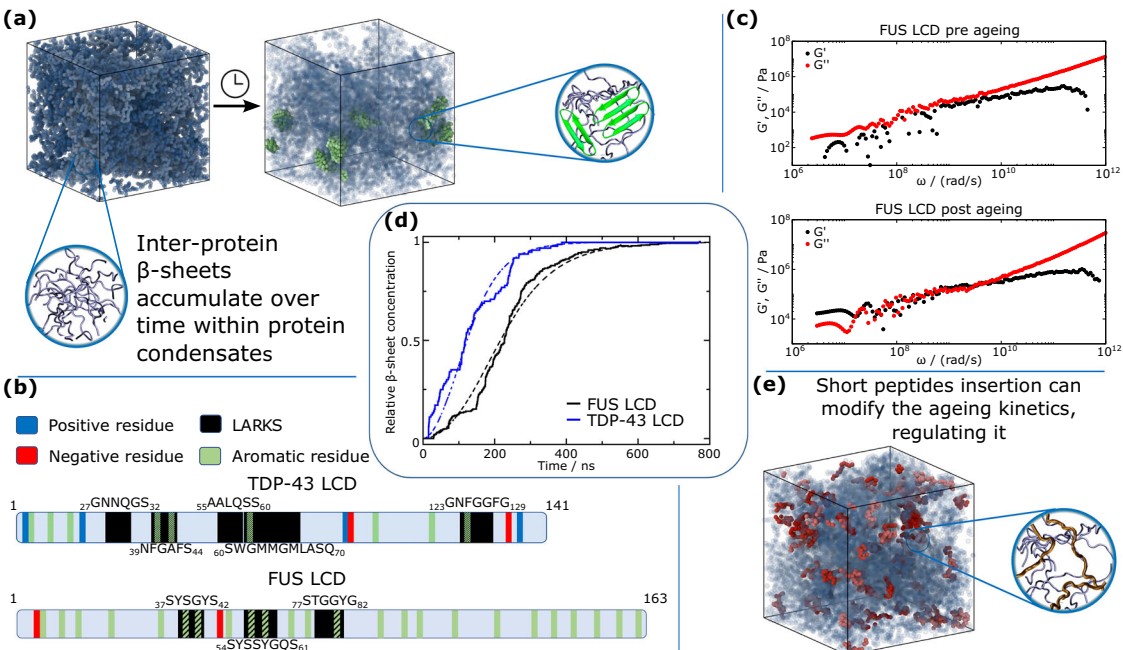

**Fig. 1 | Condensate ageing process. a** Schematic representation of the accumulation of inter-protein $\beta$-sheets within a bulk condensate simulation. The zoomed images depict the nature of the initially intrinsically disordered protein regions (left) and the cross-$\beta$-sheet structures formed over time (right). **b** TDP-43 LCD and FUS LCD sequence representation, highlighting the charged and aromatic residues, as well as the low-complexity aromatic-rich kinked segments (LARKS) as indicated in the legend. **c** Storage (G') and shear loss (G'') moduli as a function of frequency ($\omega$) for non-aged (top) and aged (bottom) FUS LCD condensates. **d** Time-evolution of cross-$\beta$-sheet concentration for FUS LCD and TDP-43 LCD condensates. The continuous lines depict simulation data while the dashed lines represent fits to the data according to a secondary nucleation dominated mechanism (see Section II C for details on these fits). **e** Simulation of a FUS LCD condensate in bulk conditions containing small peptides that hinder the nucleation of cross-$\beta$-sheet structures.

segments (LARKS) that have been found capable of forming inter-protein $\beta$-sheet structures[46,79]. By means of MD simulations, we perform a high-resolution analysis of peptide-protein interactions over time, capturing detailed dynamics and structural changes within condensates that experimental techniques are unable to detect[5,80–92]. We employ the residue-level resolution CALVADOS2 model[78] to explicitly represent each amino acid, thereby capturing essential sequence-dependent effects in condensate phase behaviour while enabling us to perform simulations over longer timescales and larger systems than would be possible with fully atomistic models[93–95]. The CALVADOS2 model[78] accurately predicts the conformational properties and propensities to undergo LLPS for diverse sequences and solution conditions[78,90] while providing fair predictions for condensate material properties despite being far beyond its parametrization scope[90].

We first investigate how the stability of FUS LCD and TDP-43 LCD condensates varies as a function of the concentration of the inserted peptides, and later we examine how the ageing rate is modulated by their recruitment into the condensate. Our goal extends from designing promising peptides that decelerate ageing to uncover the underlying mechanisms and interactions behind their inhibitory effect, which can further guide the design of functional sequences for inhibiting condensate ageing. To achieve this, we analyse inter-protein contact frequency maps, interrogate how condensate density and stability regulate ageing kinetics, and explore potential peptide-protein specific interactions that influence the hardening process. Overall, our work proposes a computational scheme to efficiently (computational benchmarks detailed in SM Section SIII) scan potential inhibitors of liquid-to-solid condensate transitions and provides insights into the underlying molecular interactions controlling the phase behaviour and time-dependent material properties of protein condensates.

## Results and discussion

### A coarse-grained modelling approach for designing peptide sequences controlling condensate stability and ageing

Our method involves a multistep approach in which we first analyse how the phase diagram of protein condensates is influenced by the amino acid composition of the designed peptides to preliminarily identify candidates that are capable of slowing down condensate ageing. It has been shown both experimentally[51,52] and computationally[50,96,97] that an increase in the density of the condensates enhances local high-density fluctuations of protein LARKS (i.e., density fluctuations in which a high concentration of multiple LARKS are found together in a small volume) which enhance the probability of inter-protein $\beta$-sheet transitions. In contrast, recruitment of partner molecules (e.g., IDRs, RNAs, or other biomolecules) which partially decrease condensate concentration reduces the probability of cross-$\beta$-sheet fibril formation[49]. To predict the variation in condensate density as a function of the concentration of the inserted peptides, we employ the CALVADOS2 residue-resolution model[78]. In this model, each amino acid across the sequence is represented by a single bead with its own chemical identity. The low-complexity domains of both FUS and TDP-43 are considered as fully flexible polymers in which subsequent amino acids are connected by harmonic bonds. The different residue-residue interactions in the model consist of a combination of electrostatic and hydrophobic interactions[78,98] implemented through a Coulomb/Debye-Hückel potential and an Ashbaugh/Hatch potential, respectively (further technical details of the model can be found in the Supplementary Material (SM) Section SI).

Once determined the condensate density (or concentration) variation upon peptide insertion through short NPT (constant number of particles (N), pressure (P) and temperature (T)) Molecular Dynamics simulations (further details on these simulations are provided in Section II), we perform non-equilibrium simulations—using our ageing dynamic algorithm[50,99]—both in the presence and absence of peptides, in which

condensates can gradually transform their material properties due to the emergence of inter-protein $\beta$-sheet structures. These simulations are performed in the canonical (NVT) ensemble at the condensate equilibrium density obtained from the NPT simulations. Within these simulations, the intermolecular interactions between the amino acids that are part of the LARKS (as those highlighted in black in Fig. 1b) can be transformed over time, mimicking the disorder-to-order transitions taking place in these protein regions[46]. Further details on this algorithm, which has been parametrized based on all-atom calculations of binding free energy differences between disordered and structured LARKS in both FUS[99] and TDP-43[96], are detailed in Sections II C and SIII of the SM. We provide the necessary files to run and replicate our ageing simulations in the Data Availability Section.

We characterize the change on the condensate material properties upon ageing in Fig. 1c by measuring the storage (G′) and loss (G″) moduli as a function of frequency ($\omega$) for FUS LCD condensates before and after the formation of cross-$\beta$-sheet structures (see Section SIV of SM for details on how to obtain these quantities). Here, the frequency ($\omega$) refers to the angular frequency of an oscillatory deformation that the system would experience in a rheological experiment. The ratio between G′ and G″ gives information regarding the fluid or gel-like behaviour of the system, where G′ values greater than G″ at low frequencies indicate gel or solid-like behaviour whilst the opposite, G″ > G′, viscoelastic behaviour. Consistently, in the non-aged condensate (Top Panel), the system displays liquid-like behaviour, characterized by the fact that the magnitude of G″ is higher than G′ at low frequencies[97]. However, once the condensate accumulates cross-$\beta$-sheet structures, and establishes long-lived inter-protein interactions (e.g., of 20-45 $k_B$T as found for FUS and TDP-43 LARKS in refs. [50,96]), elasticity dominates over flow, and therefore, the system exhibits solid-like behaviour (G′ > G″ at lower frequencies). This experimentally reported behaviour for both FUS[52,54,100] and TDP-43[79,101] among many other proteins[102] emerges from a percolated network of cross-$\beta$-sheets assemblies which restrains the protein mobility and impedes its diffusion across the condensate as recently shown by us[50]. Although our G′ and G″ values cannot be directly compared to those obtained experimentally due to the lack of explicit solvent and the coarse-grained nature of the force field, we have recently verified that the CALVADOS2 model correctly predicts experimental variations in condensate viscosity upon specific sequence mutations[103] in the low-complexity domain of hnRNPA1 protein, which also exhibits condensate ageing over time[104]. In Fig. S1, we show the time-evolution of G′ and G″ across intermediate ageing states to show their gradual crossover upon the accumulation of cross-$\beta$-sheets.

In our scheme, we analyse the kinetics of this process to determine how peptides modify the nucleation time and growth rate at which inter-protein $\beta$-sheet structures are formed. In Fig. 1d, we show the time evolution of the relative concentration of LARKS forming cross-$\beta$-sheets in pure condensates of both FUS LCD and TDP-43 LCD. These curves allow direct determination of the different parameters that serve as a metric for assessing the ageing rate. By performing multiple independent simulations, we determine the characteristic timescales of cross-$\beta$-sheet formation in each protein condensate. These parameters (detailed in Section IIC) determine whether there is an effective deceleration of the ageing kinetics as a function of the composition, patterning, and concentration of the inserted peptides with respect to the pure protein condensate. In what follows we probe this method for protein condensates in presence of small peptides, with the aim that their recruitment within the condensate frustrates the protein high-density fluctuations leading to cross-$\beta$-sheet transitions. We provide a schematic explanation in Fig. 1e of how small peptides (coloured in red) are intertwined within the protein liquid network (coloured in navy blue), and potentially inhibit the formation of inter-protein $\beta$-sheets, either by lowering the overall protein density, or by establishing selective interactions with aggregation-prone protein regions.

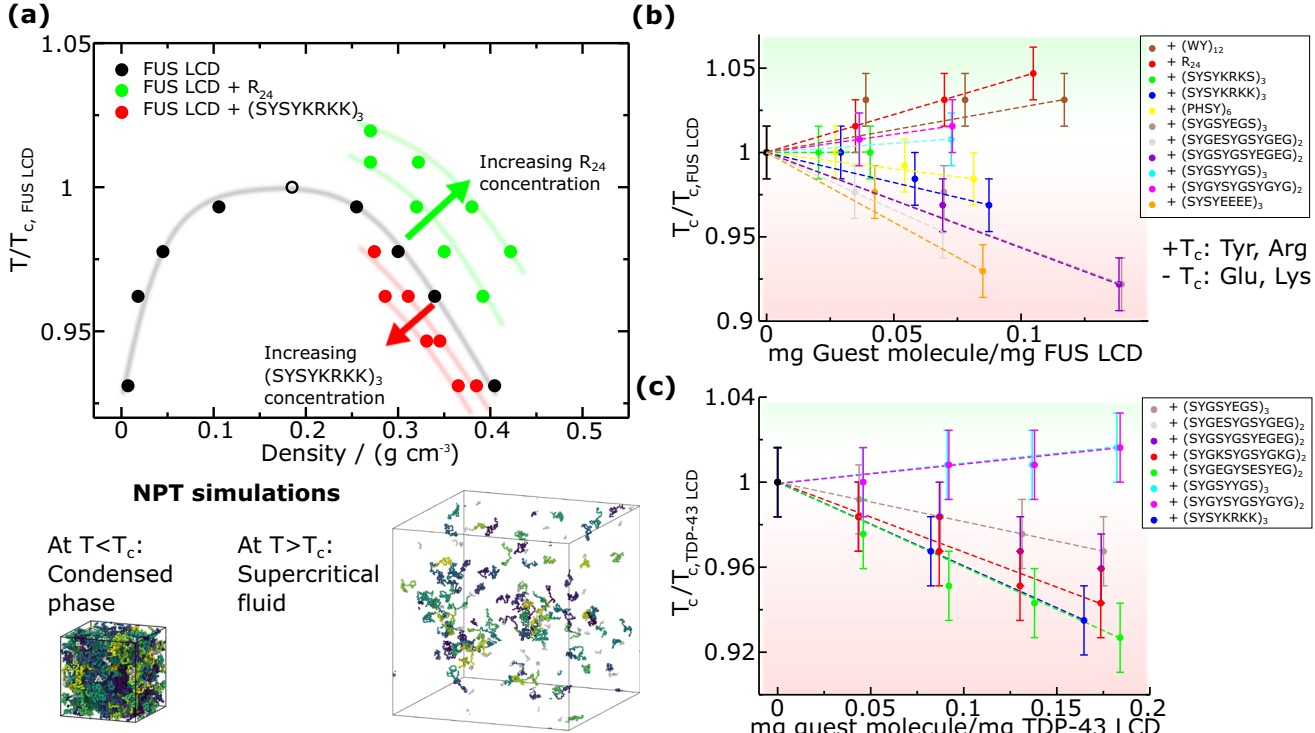

**Fig. 2 | Stability bounds of biomolecular condensates upon peptide insertion.**
**a** Normalized temperature (where $T_{c,FUS\ LCD} = 322\ K$) *vs.* density phase diagram for FUS LCD in presence and absence of two peptides modulating its phase behaviour. Increasing concentrations of $R_{24}$ augment the condensate density and critical solution temperature, while $(SYSYKRKK)_3$ induces the opposite behaviour. Filled points represent direct measurements under coexistence conditions, while the empty point depicts the critical temperature for LLPS. The continuous lines are shown as a guide for the eye. At the bottom we represent typical snapshots of the NPT simulations employed to compute the phase diagram in presence of small peptides, where at temperatures below the critical one, the condensed phase

remains stable at $P = 0$ bar, while above $T_c$, the system forms a low density phase. **b** Normalized critical temperature of FUS LCD mixtures with different peptide sequences (as indicated in the legend) as a function of the peptide concentration. **c** Normalized critical temperature (where $T_{c,TDP\text{-}43\ LCD} = 308\ K$) of TDP-43 LCD mixtures with different peptides as a function of the peptide concentration. The green and red shaded areas in (b, c) represent an increase and decrease in $T_c$, respectively. The error bars are obtained as the interval between the highest temperature at which the condensate is stable and the lowest one at which it is not, under $P = 0$ conditions.

## Phase boundaries of FUS LCD and TDP-43 LCD condensates in presence of small peptide modulators

To prevent the formation of inter-protein $\beta$-sheet structures without significantly altering the stability and material properties of the condensates, peptides must only hinder local LARKS high-density fluctuations (driving cross-$\beta$-sheet formation) while preserving the central interactions between protein domains that dictate the saturation concentration and critical solution temperature of the condensate. For that purpose, the concentration of peptides within the condensates shall be kept moderately low. We have verified that the viscoelastic properties of the condensates remain almost identical upon addition of small peptides in moderate concentrations. In Fig. S2 we show G(t), and calculate the viscosity for FUS LCD in absence vs. presence of a short peptide $((SYGSYEGS)_3)$, proving how it remains almost identical. We then begin by estimating the phase diagram of pure FUS LCD condensates (Fig. 2a) using Direct Coexistence (DC) simulations[5], in which the condensed and diluted protein phases coexist in the same elongated simulation box. Through the combination of DC simulations and the laws of rectilinear diameters and critical exponents[105] (further details in SM Section SV), we determine the coexistence densities and critical solution temperature for FUS LCD pure condensates (Fig. 2a; black filled and empty symbol, respectively). We define the critical solution temperature for LLPS as our metric of biomolecular condensate stability[90] when comparing between different protein/peptide systems. Since imposing a given concentration of peptides within a protein condensate is hardly attainable through DC simulations[83], we perform NPT simulations for protein/peptide mixtures, in which only

the condensed phase in bulk conditions is simulated. From these NPT simulations, the critical solution temperature ($T_c$) is estimated to be between the highest temperature at which the condensate is stable at $P = 0$ atm and the lowest one at which the condensed phase evolves into the diluted phase (Fig. 2a, bottom; further details on this method are provided in Section SII of the SM). This approach—validated in ref. 83—additionally allows to evaluate the condensate density as a function of temperature for different peptide sequences and/or protein/peptide stoichiometries (Fig. 2a; red and green symbols). Our simulations are performed at physiological NaCl concentration (150 mM), which is imposed through the Debye length of the screened Coulombic potential[78,106] (see SM section SI for further details on the coarse-grained model). Strikingly, we find that increasing concentrations of short polyArginine peptides of 24 residues ($R_{24}$) enhance the stability of FUS LCD condensates despite their highly self-repulsive character. On the other hand, addition of $(SYSYKRKK)_3$ peptides, which are also positively charged, reduces condensate density and $T_c$ upon its recruitment within the condensate (Fig. 2a). We attribute such opposite behaviour to the strong cation-$\pi$ interactions that $R_{24}$ peptides are capable of establishing with the aromatic residues of FUS LCD sequence at low concentrations; in contrast to the less energetic contacts that lysines (K) and serines (S) can form with FUS LCD.

To further investigate the impact of peptide regulation on FUS LCD phase-separation propensity, we perform a systematic study of different representative sequences that present specific variations of patterning and composition (Fig. 2b). We note that there are millions of possible peptide combinations that may be used to modulate

biomolecular condensate phase behaviour; however, we design 15 different sequences of 24 residues per chain enriched in specific amino acids that frequently appear in RNA-binding proteins (RBPs) such as FUS or TDP-43. This strategy enables us to maintain as much as possible the protein condensate composition of the pure protein system. The chosen residues are tyrosine (Y), serine (S), and glycine (G), where Y acts as the main sticker mediating $\pi$-$\pi$ interactions with aromatic-rich regions in FUS LCD and TDP-43 LCD, while S and G function primarily as neutral spacers contributing to peptide flexibility and solubility. Moreover, we also introduce charged residues across the peptide sequences: lysine (K), arginine (R), and glutamic acid (E) to generate peptide-peptide self-repulsion that partially destabilize LARKS high-density fluctuations leading to cross-$\beta$-sheet clustering. The choice of including both R and K when designing positively charged peptides is their different roles in phase-separation. Arginine is a context-dependent sticker[107] due to its capacity to form strong cation-$\pi$ interactions, whereas K mostly acts as a spacer only favouring electrostatically oppositely charged contacts[108,109]. In contrast, both E and aspartic acid (D) have been shown to play comparable roles when mediating electrostatic interactions that support intermolecular associations in condensate formation[109]. Therefore, for simplicity, we only consider E when designing negatively charged peptides.

In Fig. 2b, we show how different peptide sequences and concentrations decisively regulate $T_c$ in FUS LCD/peptide mixtures as compared to pure condensates ($T_{c, FUS\ LCD}$) at constant physiological salt concentration. The first two peptides we explore are the tryptophan (W)-tyrosine (Y) 12-repeat sequence ($(WY)_{12}$) and $R_{24}$. $(WY)_{12}$ is a purely aromatic sequence formed of 'sticker' residues[110–112] which are capable of forming strong $\pi$-$\pi$ interactions with FUS LCD. $R_{24}$ only contains arginine, and displays high binding affinity to FUS LCD aromatic residues such as Y and phenylalanine (F) through cation-$\pi$ interactions, as well as moderate repulsive interactions with other positively charged residues[109]. Interestingly, despite being $(WY)_{12}$ a purely self-attractive peptide and $R_{24}$ a self-repulsive peptide, both of them monotonically increase the critical solution temperature for phase separation across the concentration range studied here (Fig. 2b; brown and red symbols). Such increase in $T_c$ implicitly produces an increase in the density of the condensate (Fig. 2a) which implies higher local density fluctuations in FUS LCD LARKS with respect to the pure condensate, thus potentially promoting more frequent cross-$\beta$-sheet transitions and faster ageing. The LLPS enhancement observed in presence of $R_{24}$ is consistent with experimental findings[113] reporting an increased propensity of RNA-binding proteins such as TDP-43 for condensation and insolubility in presence of arginine-rich peptides (i.e., proline-arginine dipeptide repeats such as $(PR)_{25}$). Therefore, short peptides aiming to partially reduce LARKS high density fluctuations must contain less 'sticker' amino acids so the overall condensate density is moderately reduced while still enabling LLPS (Fig. 2a; red symbols). Similarly to these two peptides, we test $(SYGSYYGS)_3$ in which charged residues are excluded (cyan curve). Although this sequence contains fewer aromatic residues than $(WY)_{12}$, it still increases the critical solution temperature for phase-separation due to the absence of self-repulsive interactions that contribute to reducing condensate density, hence it will not be further considered as a potential ageing inhibitor.

We next investigate the effect of a combination of 'sticker' and 'spacer' amino acids. For this purpose, $(PHSY)_6$ peptides are added to FUS LCD condensates, where proline (P), serine (S) and histidine (H), act as spacers. Remarkably, we find how this sequence is able to lower the critical temperature in a marginal extent up to relatively high concentrations (>0.075 mg peptide/mg FUS LCD). Nevertheless, since a high concentration of $(PHSY)_6$ is required to slightly reduce the density of the condensate, this peptide is not ideal to preserve: (1) the material properties of FUS LCD condensates; and (2) the overall condensate composition, which in turn impacts other physicochemical

parameters such as the surface tension (Fig. 2b). We also test the influence of partially charged peptides on the stability of the condensates. We first insert strong negatively charged peptides to destabilize FUS LCD homotypic intermolecular interactions near the LARKS regions via electrostatic-driven binding affinity between K and E residues from FUS LCD and the peptides. $E_{24}$ (only glutamic acid, not shown in Fig. 2b) peptides destabilize the condensate by at least 0.1 $T_{c,FUS\ LCD}$ at the lowest concentration measured, making it not suitable for this purpose since 0.1 $T_{c,FUS\ LCD}$ implies a reduction of ~ 30K in the critical solution temperature.

We then design four additional peptide sequences with lower content of glutamic acid ($(SYSYEEEE)_3$, $(SYGSYEGS)_3$, $(SYGESYGSYGEG)_2$, and $(SYGSYGSYEGEG)_2$), each of them with a different patterning and composition of negatively charged residues (E), stickers (Y), and spacers (G, S). We find that $(SYSYEEEE)_3$ still destabilizes FUS LCD condensates to a great extent by lowering the critical solution temperature at significantly low concentrations (>0.05 $T_{c,FUS\ LCD}$ with respect to pure FUS LCD condensates at mass ratios <0.05 mg peptide/mg FUS LCD). While certainly a decrease in the critical temperature and density of the condensates is intended, the concentration at which that effect is achieved is too low, and therefore, we do not consider $(SYSYEEEE)_3$ as a potential cross-$\beta$-sheet inhibitor. Its strong capacity to disrupt the FUS LCD liquid network connectivity is not optimal if its relative concentration within the condensate is not at least 0.05 mg peptide/mg FUS LCD. This is because the concentration of the inserted peptides must be high enough to interact with a significant number of protein LARKS to effectively disrupt high density fluctuations over time leading to cross-$\beta$-sheet transitions. By testing the three other sequences with a lower concentration of E ($(SYGSYEGS)_3$, $(SYGESYGSYGEG)_2$ and $(SYGSYGSYEGEG)_2$), we find the desired destabilizing effect of 5-10 K in $T_c$ within a concentration range of 0.05 to 0.1 mg peptide/mg FUS LCD. Such destabilizing driving force meets a reasonable compromise between a moderate reduction in condensate density and stability while ensuring a sufficient concentration of peptides to target LCD-prone aggregation domains. We note that a similar behaviour is expected to be achieved if aspartic acid instead of E would have been sequenced across the peptides due to their physicochemical resemblance at modulating protein LLPS as reported in refs. 78,109,114,115.

Lastly, we evaluate how the presence of positively charged peptides impacts the critical solution temperature of FUS LCD. We design two sequences of peptides containing a common segment SYSY—with high associative binding for LARKS and adjacent regions[96]—and different positively charged domains (KRKS and KRKK, respectively) to display a certain degree of electrostatic self-repulsion. These peptides are: $(SYSYKRKS)_3$ and $(SYSYKRKK)_3$. Examining FUS LCD phase behaviour as a function of the peptide concentration shows that only $(SYSYKRKK)_3$ meets the criterion of inducing a moderate destabilization effect in the critical solution temperature (-0.04 $T_{c,FUS\ LCD}$) at low mass ratios (i.e., -0.075). Such difference in the phase behaviour between both peptides stems from the increased abundance of lysine in $(SYSYKRKK)_3$, which enhances electrostatic self-repulsion. Consequently, this reduces condensate density and stability, akin to the impact experimentally[116,117] and computationally[83,118] reported when high concentrations of poly-Uridine (negatively charged in this case) are recruited into RNA-binding protein condensates.

Furthermore, we have conducted a similar analysis involving TDP-43 LCD condensates (Fig. 2c). We probe the stability of the condensates in the presence of the peptide sequences found to be successful in reducing the density and stability of FUS LCD condensates: ($(SYSYKRKS)_3$, $(SYSYKRKK)_3$, $(SYGSYEGS)_3$, $(SYGESYGSYGEG)_2$ and $(SYGSYGSYEGEG)_2$). As reported in Fig. 2c, the five sequences achieve a reasonable compromise between a moderate reduction in condensate density and stability at a sufficient concentration of peptides capable of targeting multiple TDP-43 proteins at once. In addition, we test one

more peptide, (SYGKSYGSYGKG)$_2$, in which the content of lysine has been reduced. Since TDP-43 LCD is slightly positively charged (e.g., +2e across its sequence), this peptide destabilizes the condensate within the desired range of concentration (Fig. 2c; red symbols) through a combination of homotypic self-repulsion and a subtle balance of heterotypic interactions: electrostatic repulsion (mediated by positively charged residues) and $\pi$-$\pi$ attraction (induced by aromatic amino acids). The same destabilizing effect mediated via electrostatic repulsion is thought to be responsible for the RNA-driven re-entrant behaviour of RBP condensate stability[116–119]. Finally, as in FUS LCD condensates, we also evaluate the impact of recruiting non-charged peptides enriched in sticker residues: ((SYGSYYGS)$_3$ and (SYGYSYGSYGYG)$_2$). Consistent with our findings for FUS LCD, both peptide sequences monotonically enhance condensate stability across the studied range of concentrations, therefore they are unable to prevent LARKS high-density fluctuations driving condensate hardening.

Overall, from our simulations of both FUS LCD and TDP-43 LCD mixtures we conclude that only self-repulsive charged peptides with a certain degree of aromatic residues and neutral spacers enable mild variations on condensate stability while remaining at low stoichiometries (e.g., 1 peptide per every 10-20 proteins forming the condensate). Recent computational work[120] has shown how electrostatic and hydrophobic residues play a crucial role in the regulation of condensate stability through non-specific interactions, in agreement with our findings for FUS LCD and TDP-43 LCD. Moreover, computational studies have also predicted that the inclusion of arginine-rich peptides enhances the stability of protein condensates[121,122] (at least for a certain concentration range), as found in our FUS LCD/R$_{24}$ mixtures (Fig. 2b). This behaviour is influenced by the cation-$\pi$ interactions which are established between peptides and proteins, as reported through atomistic simulations in refs. 121,123. Furthermore, we note that another recent computational study has also predicted the suppression of hnRNPA1 phase separation after the addition of glutamic acid enriched peptides[124], which is in close agreement with our observations (Fig. 2b, c) and previous simulations studies of hnRNPA1 in the presence of polyUridine[50,109,118]. However, while the factors which modulate the stability of protein condensates in presence of small molecules and peptides is beginning to be understood, their impact on the condensate time-dependent material properties remains unclear.

## Charged peptides enriched in aromatic residues decelerate condensate ageing driven by inter-protein structural transitions

To elucidate the impact of peptide inclusion in condensate material properties and ageing, we now perform non-equilibrium MD simulations incorporating our ageing algorithm, which accounts for the description of structural and energetic transformation of disordered LARKS into inter-protein $\beta$-sheets[50,96]. As input for the free energy binding difference between the structured vs. disordered LARKS, we use atomistic potential-of-mean-force (PMF) calculations from refs. 96,99 for FUS LCD and TDP-43 LCD, respectively. In these studies, calculations for the dissociation of the inter-protein $\beta$-sheet clusters were evaluated based on the structures from refs. 46,79, resolved via Cryo-Electron Microscopy (Cryo-EM). PMF calculations were performed using an atom-level force field[125], considering both structured conformations of refs. 46,79 and the intrinsically disordered ensemble predicted by the all-atom model[125].

Our algorithm continuously evaluates LARKS high-density fluctuations across the simulation using a distance criterion based on a local order parameter developed by us[99]. When at least four LARKS from different protein replicas meet specific proximity conditions (i.e., within a cut-off distance consistent with the free energy minima of the atomistic calculations), the interactions between these segments are adjusted according to the structured binding free energy of the given sequence domain[50]. These new interactions become

significantly stronger (i.e., from 3-6 k$_B$T per LARKS when being disordered to 30-50 k$_B$T upon the structural transition[96]) mimicking the emergence of an inter-protein $\beta$-sheet structure (Fig. 1a). While LARKS cross-$\beta$-sheets are not usually thermoresistant, as amyloid fibrils[52,126], these structures are still highly stable as experimentally[39,46–49] and computationally[50,99,127] found. Moreover, besides the energetic implications of the structural change, our algorithm also imposes an angular potential between consecutive amino acids to reflect the increased rigidity of the newly formed structures, thus replicating the conformational change across the involved segment. For further details on the local order parameter, cut-off distances, magnitude of the high-density fluctuations, and binding energies associated to the structural transitions, please see Section SIII of SM.

We perform non-equilibrium simulations in bulk conditions (at the corresponding pre-ageing condensate density) for both FUS LCD and TDP-43 LCD condensates in the presence of the previously selected peptides at a normalized temperature of T/T$_c$ = 0.95 (293 and 305 K for TDP-43 LCD and FUS LCD, respectively). The inserted peptides are at the maximum concentration which lowers the critical temperature for phase separation in ~0.03 T$_c$ (the absolute numerical value of each concentration is specific for each system and is reported in the SM section SVII). The inter-protein $\beta$-sheet concentration as a function of time for different FUS LCD trajectories in absence vs. presence of different peptide sequences is shown in Fig. 3a. Since the nucleation of cross-$\beta$-sheet stacks is a stochastic process[128], we carry out 6 independent trajectories of each system with different initial velocity seeds. The inter-protein $\beta$-sheet concentration is normalized by the maximum number of cross-$\beta$-sheets formed (at the plateau), which approximately corresponds to 90% of the total possible $\beta$-sheets formed. As shown in Fig. S1, such high concentration of cross-$\beta$-sheets precludes the protein mobility within the condensate leading to a solid-like aggregate. While pure FUS LCD condensates (light blue curve) rapidly develop inter-protein structural transitions over time, condensates including peptides such as (SYGSYEGS)$_3$ (brown curve) display considerably longer lag times (~100 times slower) before the formation of $\beta$-sheet assemblies. The ageing curves for all the studied systems containing the different independent trajectories per system are reported in Fig. S3. According to our simulation setup, and based on the obtained trends, the process follows a nucleation-elongation mechanism[128], in which the nucleation stage is the rate-limiting step of the process, and precedes the elongation in which many more cross-$\beta$-sheets grow rapidly. Notably, until the first structured assembly is formed, proteins display liquid-like behaviour characterized by transient and weak interactions. However, once a stable nucleus emerges, the elongation step rapidly evolves, giving rise to a sharp increase in the cross-$\beta$-sheet concentration (Fig. 3a). Remarkably, we find that fitting our results to a secondary nucleation-dominated kinetic model[129] (dashed curves fitted with Amylofit software[130]) matches the simulation data. Hence, our simulations suggest that both cross-$\beta$-sheet growth and secondary nucleation represent two fundamental mechanisms by which kinetically arrested states are reached.

To further characterize the ageing kinetics, we evaluate the time required to reach half of the cross-$\beta$-sheets formed in the condensate (hereafter referred to as the half-time, or t$_{1/2}$). t$_{1/2}$ is a standard metric for quantifying the kinetics of aggregation from condensates and/or supersaturated protein solutions[69,72,131–133]. In Fig. 3b, we report the box plots for the average half-times (from independent trajectories) obtained for the different peptide/FUS LCD mixtures. The average value is indicated with a cross symbol, the boxes represent the 25-75 quartile range, and the error bars indicate the highest and lowest values obtained. We conclude that ageing kinetics are substantially decelerated in the presence of most of the chosen peptides selected based on the density-stability criterion discussed in Section II B. In order to determine whether the deceleration in the cross-$\beta$-sheet formation is statistically significant, we perform a Mann-Whitney U test

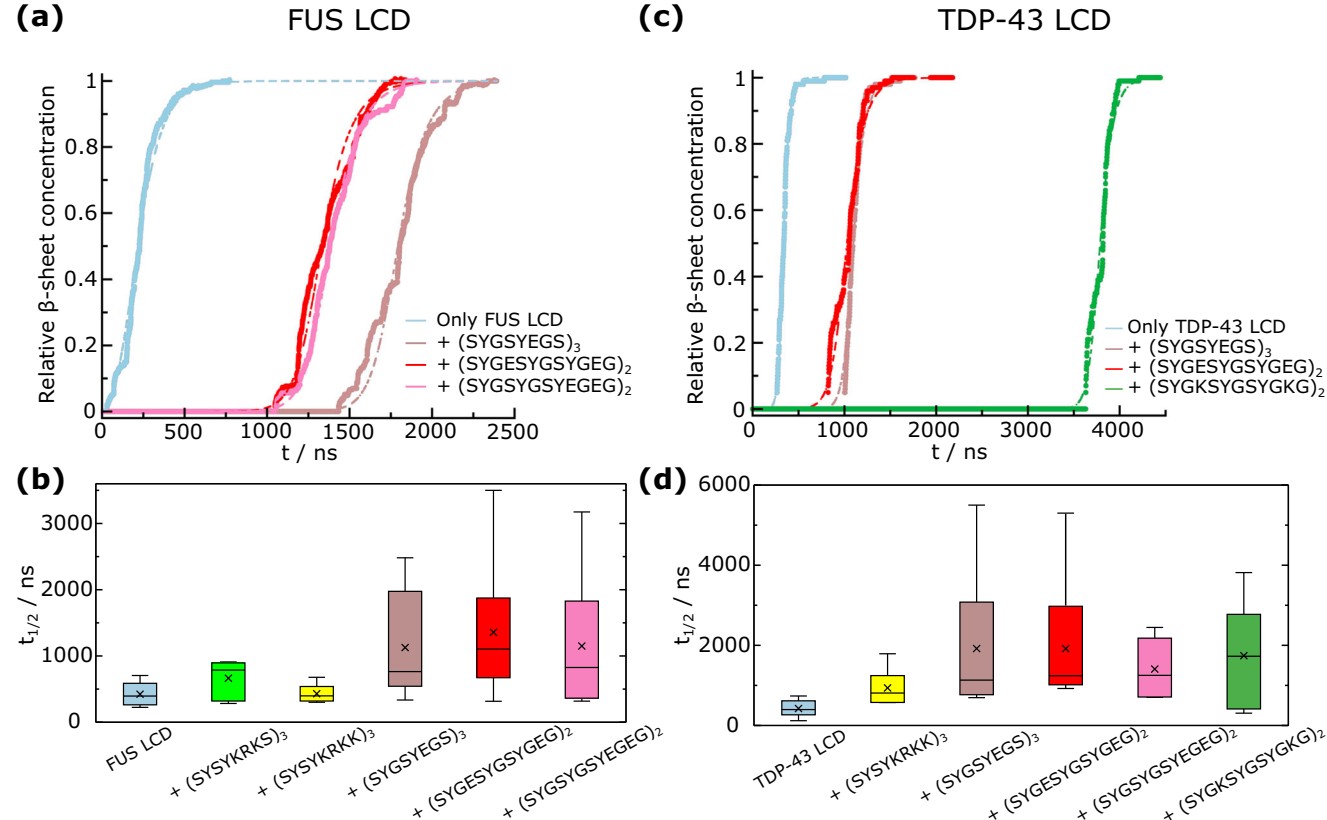

**Fig. 3 | Ageing kinetics is controlled by the presence of small peptides.**
**a** Relative cross-$\beta$-sheet concentration—determined from the amount of inter-protein $\beta$-sheets formed—as a function of time for FUS LCD condensates in absence vs. presence of different peptide sequences as indicated in the legend. Filled points represent measured $\beta$-sheet concentration from non-equilibrium simulations, while dashed lines depict fits to a secondary nucleation dominated kinetic model performed with Amylofit[130]. We present only 3 representative trajectories that illustrate the peptide-mediated ageing deceleration. **b** Half-time ($t_{1/2}$) for the different FUS LCD condensate mixtures studied. $t_{1/2}$ is defined as the point where the relative $\beta$-sheet concentration value is halfway between the initial value and final plateau value. In the box plot representations, the average value is represented with a cross symbol, the boxes represent the 25−75 quartile range (with the horizontal bar being the median), and the whiskers indicate the highest and lowest values. **c** Relative cross-$\beta$-sheet concentration as a function of time for TDP-43 LCD condensates in absence vs. presence of different peptide sequences as indicated in the legend. Filled points and dashed curves represent the same quantities as in (**a**). **d** Half-time for the different TDP-43 LCD mixtures studied. The average value is represented with a cross symbol, the boxes represent the 25−75 quartile range (with the horizontal bar being the median), and the whiskers indicate the highest and lowest values.

(numerical results reported in SM Section SVIII and Table S3). For all the tested sequences, only (SYSYKRKK)$_3$ is not capable of significantly deferring the nucleation time. On the other hand, (SYGSYEGS)$_3$, (SYGESYGSYGEG)$_2$, (SYGSYGSYEGEG)$_2$, and (SYSYKRKS)$_3$ (in a milder extent) can substantially disrupt LARKS high-density fluctuations triggering FUS LCD progressive solidification. Nevertheless, if the concentration of peptides (like (SYSYKRKS)$_3$ which only provokes a moderate deceleration on ageing kinetics) is doubled, a significant deceleration in the cross-$\beta$-sheet formation can be observed (Fig. S13). However, such high concentration of peptides, beyond modulating ageing dynamics, also strongly influence condensate stability as shown in Fig. 2b.

In FUS LCD, we find that negatively charged peptides perform significantly better than positively charged ones in increasing $t_{1/2}$. The FUS LCD sequence only contains 2 negatively charged residues out of 163 amino acids, nevertheless, such additional degree of destabilization through electrostatic repulsive forces that negatively charged peptides induce in FUS LCD condensates contributes to further hindering condensate ageing. We also find that including peptides with the same amino acid composition but different sequence patterning, such as (SYGESYGSYGEG)$_2$ and (SYGSYGSYEGEG)$_2$, appears to play a moderate role in modulating the onset of cross-$\beta$-sheet formation in FUS LCD. Similarly, the change in E abundance between (SYGSYEGS)$_3$ and both (SYGESYGSYGEG)$_2$ and (SYGSYGSYEGEG)$_2$ does not show a

major impact in varying the ageing propensity as long as the balance between peptide self-repulsion and protein binding affinity for being recruited into the condensate is maintained. In Section II D, we will perform a molecular contact analysis of the condensate mixtures to uncover which major intermolecular interactions are responsible for the deceleration of cross-$\beta$-sheet formation.

From the box plot analysis, we also reveal that ageing kinetics is governed by the nucleation lag time, which initially prevents the rigidification cascade from occurring. Since the nucleation step is a stochastic event that takes place with a given probability (depending on the condensate composition, density, and thermodynamic conditions), it is expected that the median (horizontal line) is lower than the average (black cross), and the distribution is wider for a lower probability of nucleation—as it happens in presence of negatively charged peptides. The $t_{1/2}$ data conforming the box plots shown in Fig. 3b follow a Poisson distribution[134], which also reaffirms the role of the nucleation stage as the crucial limiting step in condensate ageing driven by cross-$\beta$-sheet ordering. We note that apart from the ageing half-time, the nucleation lag time (defined as the time at which the first inter-protein $\beta$-sheet is formed), and -log($k_n$) (where $k_n$ is the primary nucleation rate constant[130]) can also be used to quantify the extent to which the ageing rate is altered by the presence of peptides. These additional analyses showing how the nucleation time, log($k_n$) and $t_{1/2}$ provide equivalent conclusions for the ageing kinetics of

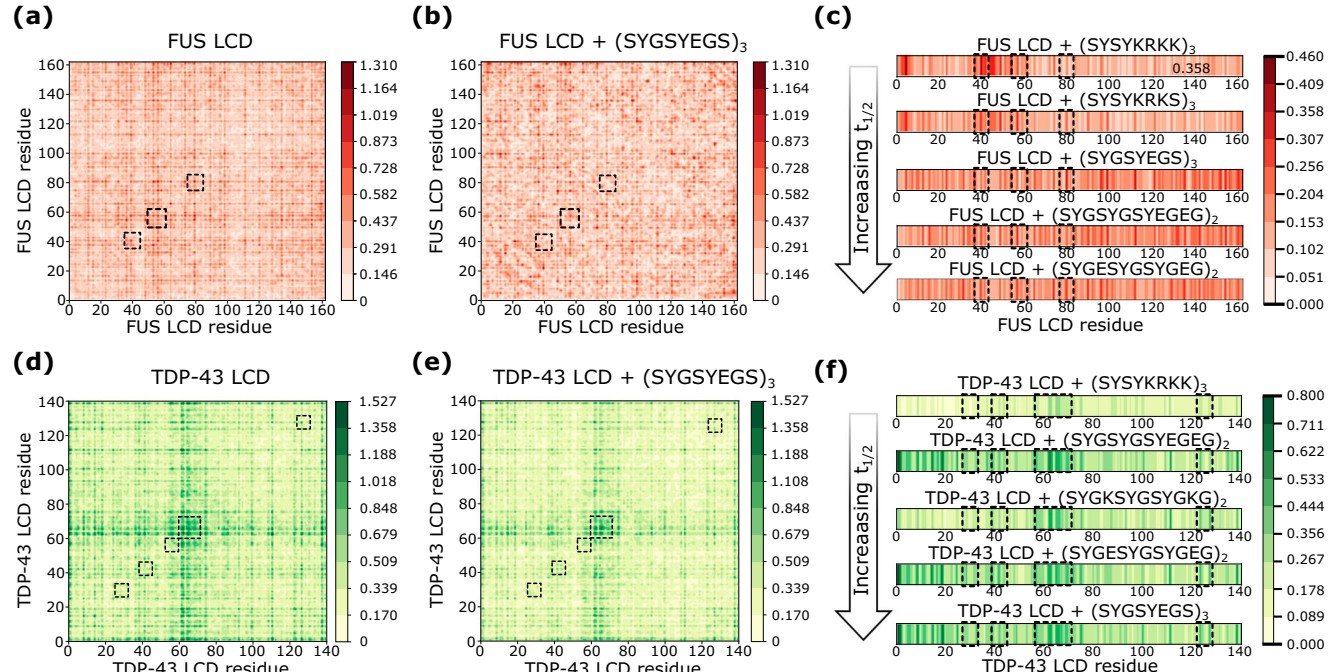

**Fig. 4 | Intermolecular interactions determining the ageing kinetic modulation.** Intermolecular frequency contact maps for FUS LCD homotypic contacts in pure FUS LCD condensates (**a**) and FUS LCD/(SYGSYEGS)₃ mixtures (**b**). The pairwise residue-residue contact frequency is expressed in percentage in all panels. The homotypic LARKS-LARKS interactions are highlighted with a dashed box. **c** Intermolecular frequency contact maps of the different inserted peptides with the FUS LCD sequence in the condensate. The contacts are calculated between all residues of the inserted peptide and the FUS LCD sequence. Intermolecular frequency contact maps for TDP-43 LCD homotypic contacts in pure TDP-43 LCD condensates (**d**) and TDP-43 LCD/(SYGSYEGS)₃ mixtures (**e**). The homotypic LARKS-LARKS interactions are highlighted with a dashed box. **f** Intermolecular frequency contact maps of the different inserted peptides with TDP-43 LCD. In panels (**c**) and (**f**), the LARKS of each protein sequence are highlighted with dashed squares.

these systems are detailed in Section SVIII of the SM (Figs. S5a and S6a).

Next, we perform non-equilibrium simulations of TDP-43 LCD condensates in the absence and presence of the peptides that partially destabilized LLPS in Fig. 2c. The time-evolution of the cross-$\beta$-sheet concentration for multiple TDP-43 LCD mixtures is shown in Fig. 3c. We consistently observe an increase in the nucleation time when small peptides are added compared to the pure protein condensate. However, the growth of cross-$\beta$-sheet clusters occurs on a similar rate across all systems. From these curves, along with the collective analysis of the different independent trajectories for each system (Fig. S4), we conclude that the ageing kinetics is limited by the nucleation and dominated by the growth, as found for FUS LCD systems in Fig. 3a, b. To quantify the effectiveness of the selected peptides in slowing down ageing, we calculate the ageing half-time. In Fig. 3d, $t_{1/2}$ is displayed in box plots for all systems. Impressively, all chosen peptides significantly decelerate ageing; with multiple trajectories showing half-times up to two orders of magnitude larger than that of pure TDP-43 LCD condensates, particularly when incorporating the (SYGSYEGS)₃ and (SYGESYGSYGEG)₂ sequences. Similarly to FUS LCD, we perform a Mann-Whitney $U$ test, and determine that all the systems induce a significant deceleration on the ageing kinetics based on an analysis which measures statistically significant differences between the distributions of independent samples (see Section SVIII of the SM). In contrast to FUS LCD condensates, both positively and negatively charged peptides delay the nucleation stage similarly: the box plots for $t_{1/2}$ in the presence of (SYGESYGSYGEG)₂ and (SYGKSYGSYGKG)₂—where E and K residues are swapped—overlap within the uncertainty. Nevertheless, when arginine residues instead of lysines are included in the peptide, as in (SYSYKRKK)₃, the ability to hinder inter-protein structural transitions is significantly reduced. We also evaluate the role of residue patterning by inserting (SYGESYGSYGEG)₂ and

(SYGSYGSYEGEG)₂ peptides. These two peptides differ in their degree of blockiness of E and Y residues. While (SYGSYGSYEGEG)₂ display more defined negative and aromatic residue patches, (SYGESYGSYGEG)₂ possess an alternating positioning of E vs. Y across its sequence. In contrast to FUS LCD condensates where both decelerate ageing in a similar extent, the (SYGESYGSYGEG)₂ sequence enables longer lag times before cross-$\beta$-sheet fibril formation than (SYGSYGSYEGEG)₂ in TDP-43 LCD condensates. Hence, patterning variations are likely to be significant depending on the specific sequence features of the protein undergoing condensate ageing. In that sense, the degree of clustering of aromatic vs. charged residues in the peptides can remarkably influence their impact on ageing kinetics, as shown in Fig. 3d. We also evaluate the primary nucleation rate constant ($k_n$) and the average nucleation lag time in Figs. S5b and S6b. As for FUS LCD mixtures, both analysis yield the same conclusions as through the analysis of $t_{1/2}$: the addition of partially−positively or negatively−charged peptides enriched in aromatic residues effectively delay the emergence of cross-$\beta$-sheet structures (Fig. 3d).

## Molecular factors governing ageing deceleration via peptide insertion

To decipher the mechanisms by which peptide insertion delays condensate ageing, we compute the intermolecular contact frequency maps for the different systems studied in Section II C before inter-protein $\beta$-sheets are formed. We consider a residue-residue contact to effectively contribute to condensate stability when two amino acids are found at a distance below 1.2$\sigma_{ij}$, being $\sigma_{ij}$ the average molecular diameter of the two residues $i$ and $j$, and 1.2 a slightly larger distance to the minimum in the interaction potential (−1.12$\sigma_{ij}$) between the $i$th and $j$th amino acids (further details on these calculations are described in Section SIX of the SM). In Fig. 4a, b, we show the contact frequency of all the different residues for FUS LCD condensates in the absence (a)

vs. presence of (SYGSYEGS)$_3$ peptides (b). From the comparison of both maps, we find that the overall intermolecular contact frequency between FUS LCD proteins is marginally altered upon peptide insertion. However, some of the residues that establish more prominent interactions, such as the subdomains that range between the 36th to the 40th residue, from the 53rd to the 58th, and from the 77th to the 82nd, which correspond to LARKS and LARKS adjacent segments, reduce their contact frequency when (SYGSYEGS)$_3$ peptides are recruited. In Fig. S7 we report the contact frequency maps for the rest of inserted peptides, which are relatively similar to that shown in Fig. 4b, i.e., the overall inter-protein contact valency remains similar to that in pure FUS LCD condensates, but contacts between LARKS and LARKS adjacent regions are reduced.

Furthermore, we evaluate the contact frequencies between the inserted peptides and FUS LCD proteins (Fig. 4c). The three LARKS capable of forming inter-protein $\beta$-sheet structures are highlighted by black dashed squares. We find that positively charged peptides ((SYSYKRKS)$_3$ and (SYSYKRKK)$_3$) show a very strong binding affinity for the two first LARKS of the sequence and the negatively charged residues located at its N-terminal domain. However, they do not display specific binding affinity for other subregions of the sequence. Conversely, negatively charged peptides ((SYGSYEGS)$_3$, (SYGESYGSYGEG)$_2$, and (SYGSYGSYEGEG)$_2$) exhibit more homogeneous cross-interactions throughout the protein sequence, except for the negatively charged residues in the N-terminal domain. This suggests that, even for a sequence with only two negatively charged residues and no positively charged amino acids like FUS LCD, a potential strategy to preserve liquid-like behaviour is to insert charged molecules of the same sign. These peptides can establish frequent interactions throughout the sequence (e.g., as $\pi$-$\pi$ contacts) while enhancing electrostatic self-repulsion and partially disrupt inter-protein connectivity. Based on our results from Figs. 3b and 4a–c, we find that a balance of moderate electrostatic self-repulsion and non-specific attractive interactions—i.e., mediated by $\pi$-$\pi$ (or cation-$\pi$) interactions—contributes to decreasing LARKS high-density fluctuations triggering cross-$\beta$-sheet stacking. Importantly, the addition of the aforementioned peptides reduces protein condensate concentration by ~5% while largely maintaining their overall stability (Fig. 2b).

It is also remarkable that negatively charged peptides are capable of interacting in higher extent with the $_{77}$STGGYG$_{82}$ LARKS, while positively charged sequences ((SYSYKRKS)$_3$ and (SYSYKRKK)$_3$) display much stronger binding affinity for the first two LARKS of the sequence ($_{37}$SYSGYS$_{42}$ and $_{54}$SYSSYGQS$_{61}$). According to our residue-level simulations, this is a cooperative effect due to the presence of the N-terminal negatively charged residues near the location of these two LARKS, which establish attractive electrostatic interactions that further strengthen $\pi$-$\pi$ contacts among the peptides and the adjacent LARKS. Nevertheless, we note that while positively charged peptides engage more locally with the first two LARKS of the sequence compared to negatively charged peptides, their impact on ageing kinetics is moderately lower (Fig. 3b).

Furthermore, we quantify whether there is any major conformational change in FUS LCD proteins when peptides are recruited into the condensate. To that goal, we evaluate the FUS LCD radius of gyration ($R_g$) in condensate bulk conditions. Interestingly, we do not find substantial changes in $R_g$ when different peptide sequences are added with respect to the pure protein condensate (Fig. S9). Hence, a mechanism in which peptide recruitment decelerates ageing by inducing a conformational switch in FUS LCD favouring intra-molecular contacts vs. inter-molecular interactions is discarded for these systems. Moreover, we have computed the diffusion coefficient ($D$) of FUS LCD within the condensates upon peptide insertion (Fig. S11). We observe no clear correlation between slower $D$ and increased $t_{1/2}$, suggesting that moderate changes in the protein self-diffusion are not the primary factor governing the ageing kinetics. Instead, the observed

deceleration of ageing is more consistent with a reduction in condensate density driven by electrostatic self-repulsion introduced by the peptides, which likely reduces the frequency of interactions critical to the ageing mechanism.

We also examine the impact on ageing kinetics when peptides increase condensate density rather than reduce it. To that aim, we select (WY)$_{12}$, which is not self-repulsive and it significantly enhances condensate stability and protein concentration inside the condensate upon its recruitment (Fig. 2b). For this mixture, we observe a modest deceleration in the nucleation time (Fig. S12), which is comparable to the case of FUS LCD + (SYSYKRKS)$_3$. Strikingly, in FUS LCD/(WY)$_{12}$ condensates, the $R_g$ of FUS decreases with respect to the previous systems (Fig. S12), suggesting that a conformational change resulting into a more compact configurational ensemble can also modulate the ageing rate. As a consequence of such closer conformations, the ratio of intra-molecular contacts vs. inter-molecular ones becomes higher than in FUS LCD pure condensates. This result shows how an alternative mechanism might also regulate ageing kinetics in protein condensates. Nevertheless, the highest deceleration rate in cross-$\beta$-sheet formation is observed when peptides partially reduce condensate protein density instead of promoting more compact LCD conformations.

Next, we evaluate the contact frequency maps for TDP-43 LCD mixtures before ageing occurs. The inter-protein TDP-43 LCD contact maps in the absence (d) vs. presence of (SYGSYEGS)$_3$ peptides (e) are shown in Fig. 4. We find in this case that the overall inter-protein connectivity is reduced upon peptide addition as a consequence of the partial density reduction driven by self-repulsive electrostatic interactions among peptides. Crucially, the most frequent contacts, which occur between residues 61-65, that corresponds to the largest LARKS in TDP-43 (Fig. 1b), are significantly reduced by ~40%. Conversely to FUS LCD mixtures, where only minor subdomains reduce their intermolecular connectivity in advantage to other subregions which increase their connectivity (Fig. 4a, b), in TDP-43 LCD mixtures we observe a broad diminishment of inter-protein contacts across the entire sequence. In Fig. S8, we show the contact frequency maps for TDP-43 LCD in presence of the other inserted peptides, which notably resemble to that shown for (SYGSYEGS)$_3$.

We plot the intermolecular contacts of TDP-43 LCD with all peptides that decelerate ageing kinetics (Fig. 3c) as different panels in Fig. 4f. TDP-43 LCD possesses an overall positive charge of +2e with two negatively charged amino acids (1D and 1E) and four positively charged residues (3R and 1K). In TDP-43 LCD condensates, (SYSYKRKK)$_3$ insertion produces the lowest ageing deceleration of the whole set since the cross-interactions among proteins and peptides are only partially localized across the LARKS domain positioned from the 60th to the 70th residue, while the rest of the sequence mildly establishes substantial interactions with the peptides (Fig. 4f; first panel). On the other hand, (SYGKSYGSYGKG)$_2$, which contains less than half of the positive charges compared to (SYSYKRKK)$_3$, is able to foster higher binding connectivity to TDP-43 LCD, and as a consequence, this sequence is one of the most effective in retarding ageing. Our results evidence how peptides with stronger affinity for the target protein possess much greater effectiveness in frustrating cross-$\beta$-sheet formation. Ageing deceleration relies on the affinity of protein-peptide interactions, and the partial condensate density reduction induced by peptide self-repulsion. Importantly, the inserted negatively charged peptides further support these observations ((SYGSYEGS)$_3$, (SYGESYGSYGEG)$_2$, and (SYGSYGSYEGEG)$_2$) since their strong impact on the cross-$\beta$-sheet nucleation time is triggered by the prominent heterotypic protein-peptide interactions established across the sequence, with most long-lived contacts taking place in the LARKS and near-LARKS regions: $_{27}$GNNQGS$_{32}$, $_{39}$NFGAFS$_{44}$, $_{55}$AALQSS$_{60}$, and $_{60}$SWGAAGALASQ$_{70}$. In that sense, the fact that more widespread peptide-protein contacts accumulate within (or around) the LARKS in

TDP-43 LCD mixtures compared to FUS LCD condensates may explain why $t_{1/2}$ increases in greater extent in TDP-43 LCD with respect to FUS LCD (Fig. 3). Nevertheless, the nucleation lag time in FUS LCD mixtures can still be delayed up to a factor of 50 even when peptides display broad binding affinity for the entire sequence (Fig. 3c). We have also quantified if major conformational changes occur in TDP-43 LCD when peptides are added into the condensate. As found in FUS LCD mixtures, no significant changes in $R_g$ are observed upon the addition of peptide sequences which partially reduce condensate stability and density (Fig. S10). Likewise, we compute the diffusion coefficient of TDP-43 LCD for the different mixtures, and no correlation has been detected between $D$ and $t_{1/2}$ (Fig. S11).

Overall, based on our multiple computational analyses, we conclude that a combination of factors are involved in preventing the nucleation stage of cross-$\beta$-sheet structures: (1) A decrease of condensate density upon peptide insertion, which is associated to a lower critical solution temperature for phase-separation (Fig. 2); and (2) Substantial binding affinity between peptides and aromatic-rich protein domains (Fig. 4c–f). Our simulations of FUS LCD and TDP-43 LCD condensate mixtures demonstrate how electrostatic repulsion between peptides and proteins is needed for decreasing condensate density (Fig. 2b). However, when peptides exhibit excessively strong repulsion toward one another, they cannot effectively prevent LARKS high-density fluctuations since only at low concentration maintain LLPS, and at those conditions multiple condensate regions lack effective protein-peptide interactions. If peptides specifically bind to protein LARKS whilst promoting electrostatically-driven low-density regions, structural transitions become highly frustrated (Fig. 4f).

## Conclusions

In this study, we provide a detailed exploration of the key molecular parameters that enable small peptides to decelerate the ageing kinetics of protein condensates. By combining equilibrium and non-equilibrium simulations of a residue-resolution protein model[78], we examine the potential that multiple peptide sequences possess in modulating the phase diagram, ageing kinetics, diffusion coefficient, conformational ensemble, and intermolecular interactions of two protein low-complexity domains (TDP-43 LCD and FUS LCD) whose liquid-to-solid transition has been associated to several neurodegenerative disorders[25,36,101]. Our simulations reveal that an effective strategy for decelerating ageing consists of partially reducing protein condensate density while minimally altering the stability of the condensate. That can be achieved through the inclusion of small peptides—24 amino acids long in this study—which combine aromatic residues acting as 'stickers', polar and uncharged residues behaving as 'spacers', and charged residues of the same sign enforcing peptide electrostatic self-repulsion. The balance between stickers, spacers, and charged residues is critical to ensure binding affinity to the protein aromatic regions susceptible to undergo cross-$\beta$-sheet transitions, and peptide self-repulsion to lower the condensate local density near the protein LARKS. We discover that peptides containing 10–15% of charged residues—either negatively charged, e.g., D or E, or positively charged, K—exhibit electrostatic self-repulsion and reduce condensate density without heavily compromising its stability (Fig. 2b, c). We also reveal that arginine, despite being positively charged, acts as a sticker since the target binding domains for the peptides are rich in aromatic residues, and it establishes cation-$\pi$ interactions instead of contributing to condensate destabilization via electrostatic self-repulsion. By lowering the density of the condensates in ~5–10%, these peptides strongly reduce the probability of disordered LARKS transitioning into cross-$\beta$-sheets, a hallmark of condensate ageing and pathological aggregation.

We characterize the peptide-driven modulation of condensate ageing by measuring the half-time of cross-$\beta$-sheet formation (which reflects the duration of liquid-like behaviour within the condensates before turning into kinetically arrested assemblies; Fig. 3), and the

nucleation lag-time (which corresponds to the onset before the emergence of the first cross-$\beta$-sheet nucleus; Fig. S5). We find that most of the selected peptides, based on the condensate stability criterion (Fig. 2), are capable of substantially delaying the nucleation lag-time. Such deceleration in the nucleation stage is essential, as it is the rate-limiting step in solidification kinetics given that the cross-$\beta$-sheet growth barely varies upon peptide inclusion (Fig. 3a–c). Our findings for TDP-43 LCD condensates (Fig. 4f) reveal that peptides with strong binding affinity for LARKS and adjacent regions are particularly effective in delaying the nucleation lag-time, reducing the likelihood of LARKS high-density fluctuations. Remarkably, we find that these peptides also preserve the diffusion coefficient of proteins in non-aged pure condensates, and their average conformational ensemble, quantified through their radius of gyration under condensed phase conditions. The resemblance of the protein-peptide mixtures to pure protein systems in terms of critical solution temperature, protein diffusion, and protein conformational ensemble suggests that peptides can be ideal molecular regulators of pathological liquid-to-solid transitions in biomolecular condensates. Moreover, our simulations show how small sequence variations in terms of composition, amino acid patterning, and net peptide charge radically determine the impact of peptide inclusion on condensate ageing kinetics.

Taken together, our work highlights three primary factors that contribute to a peptide's inhibitory potential of condensate hardening: (1) a reduction in condensate density without altering condensate stability and protein mobility; (2) moderate but consistent peptide-protein binding affinity (i.e., driven by $\pi$-$\pi$ or cation-$\pi$ interactions) that partially disrupts inter-protein molecular connectivity; and (3) selective interactions between peptides and aromatic-rich regions (e.g., LARKS), that further destabilize LARKS-rich environments through self-repulsive interactions. These, a priori, intuitive conclusions have been reached thanks to the possibility of exploring through realistic and efficient computational models how a large set of peptide sequences govern the stability and ageing kinetics of protein condensates. Experimentally testing such amount of sequence variations can be extremely challenging[102]. However, our work proposes an efficient computational pipeline by which the impact of sequence variations can be easily tested in condensate stability, viscoelastic properties, and ageing kinetics. Extending the sampling of the studied sequences to a considerable larger set can further contribute to decelerate, or even fully inhibit, the emergence of cross-$\beta$-sheet transitions in protein condensates. In that sense machine-learning algorithms can be extremely helpful to optimize sequence iterations which maximize the nucleation lag-time of cross-$\beta$-sheet aggregation. Therefore, our study provides a promising proof-of-concept for further investigation of potential sequences as modulators of condensate-related pathologies, offering insights that may guide future therapeutic designs. Refining peptide composition to enhance the specificity of interactions with LARKS and other key protein regions, can potentially offer even more robust strategies to control biocondensate phase behaviour and their strong link to health and disease.

## Methods

The full methodological details can be found in the Supplementary Material.

### Reporting summary

Further information on research design is available in the Nature Portfolio Reporting Summary linked to this article.

## Data availability

The data that supports the findings of this study are available within the article and its Supplementary Material. Source data are provided with this paper.

## Code availability

The necessary configuration and LAMMPS script to run a non-equilibrium ageing simulation can be found in the repository https://doi.org/10.5281/zenodo.15168499. Protein sequences are reported in SM section SXIV.

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

## Acknowledgements

I. S.-B. acknowledges funding from Derek Brewer scholarship of Emmanuel College and EPSRC Doctoral Training Programme studentship, number EP/T517847/1, Ramon y Cajal fellowship (awarded to J.R.E.), as well as the UKRI EPSRC under the UK Government's guarantee scheme (EP/Z002028/1), following successful evaluation by the ERC (Consolidator Grant awarded to R.C.G.) under the European Union's Horizon Europe research and innovation programme. A. R. T. acknowledges funding from the European Union Horizon 2020 research and innovation programme (grant agreement 803326 to R.C.-G.). J.R.E. acknowledges funding from Emmanuel College, the University of Cambridge, the Ramon y Cajal fellowship (RYC2021-030937-I), and the Spanish scientific plan and committee for research reference PID2022-136919NA-C33. J.R.E. has received funding from the European Research Council (ERC) under the European Union's Horizon Europe research and innovation programme (grant agreement No 101160499). This work has been performed using resources provided by the Cambridge Tier-2 system operated by the University of Cambridge Research Computing Service (http://www.hpc.cam.ac.uk) funded by EPSRC Tier-2 capital grant EP/P020259/1-CS170. This work has also been performed using resources provided by Archer2 (https://www.archer2.ac.uk/) funded by EPSRC Tier-2 capital grant EP/P020259/e829. The authors also thankfully acknowledge RES computational resources provided by Mare Nostrum 5 through the activities 2024-3-0001 and 2025-1-0008.

## Author contributions

I.S-B. and J.R.E. conceived the project, and together with A.R.T., A.C., and A.F. carried it out; I.S.-B. designed and carried out the data analysis; J.R.E. and R.C.-G. supervised the project and provided computational means. I.S.-B. wrote the paper and all authors contributed equally to revising it.

## Competing interests

The authors declare no competing interests.
