## [Transparent Peer Review file · Nature Communications]

Charged peptides enriched in aromatic residues decelerate condensate ageing driven by cross- β -sheet formation

Corresponding Author: Dr Jorge Espinosa

Version 0:

Reviewer comments:

Reviewer #1

(Remarks to the Author)

This research presents a computational workflow to assess the impact of short peptide insertion in biomolecular condensates to mitigate their aging process (transitioning from a liquid to a solid-like state). This mechanism is associated with multiple neurodegenerative diseases, making the subject addressed here particularly important. Furthermore, the simulations elucidate the molecular principles governing the activity of these peptides, which can inspire novel designs and experiments in the future.

However, I have some concerns in relation to the time delay calculations of the ageing process. As the data is shown now, I don't think it supports the statement in the abstract: "small peptides (...) decelerate ageing up to two orders of magnitude". The authors find some changes in their averages, but I am not sure how significant they are due to Poisson distribution they follow. How much do the peptides reduce the probability of presenting a rapid aging transition? Figure 3 a and c show clear trends, but it is not clear which data is presented there (one selected curve or averaging all curves?). In addition, the authors say they run 6 replicas per system, but I only find 5 in some systems (e.g. Figure S1e).

In addition, could the authors support their findings with some experimental data? I understand that direct comparison is very difficult, but maybe there are some experiments available in the literature that produce indirect support to their findings? How do experimental G' and G'' modules compare with their simulations?

My other criticism is that I found the presentation of the methodology very confusing and cryptic at some points, so it is very difficult for me to evaluate. This is unfortunate because the authors consider their methodology as one of the assets of their manuscript. If I tried to follow up their pipeline, I would have a myriad of questions. In the SM should be clearly explained how the different type of simulations is set up, in which conditions are performed and how many replicas (ie force-field, solvent model, periodic boundary conditions, number of molecules, density and/or concentration etc..). I would also find necessary to share all their scripts so that the community could really benefit from their computational pipeline.

Another minor criticism is that I found the writing convoluted and repetitive. Each result section starts with a kind of description of the whole paper, making the text unnecessarily long. I found section IIA particularly confusing as it aims to provide a general overview but is quickly plagued with details just superficially explained. The PMF from all-atom simulations is an illustration of this point: it appears several times in the main text, but it is not till the nth occasion that it is well explained, the first mentions being totally puzzling.

I would also appreciate a more detailed explanation of how the authors decided the different patterns for peptide sequences.

Finally, the authors claim that their methodology is "rapid," although they do not offer any quantification. How long do their simulations take?

List of detailed comments:

- "Local high-density fluctuations": This expression appears in several places, but it is not clear to me what it means.
- NpT is not defined the first time it appears in the text. Why the authors think it is worth specifying that these simulations are N, P, T constant. Aren't they all? Why is 'p' not a capital letter?

- If I understood well, the NpT simulations go first in the computational pipeline, but then they are described after the 'aging simulations' in SM. Why? I found this really confusing.
- How are G' and G'' calculated, and why are they obtained as a function of frequency? Frequency of what?
- What is the difference between DC and NpT simulations?
- It is not clear how the critical temperature is calculated, whether using DC or NpT simulations
- “ Through the combination of DC simulations and the laws of rectilinear diameters and critical exponents [103], we determine the coexistence densities and critical solution temperature for FUS LCD pure condensates (Fig. 2(a); black filled and empty symbols, respectively).” This should be explained in the SM.
- How the authors establish which are the good criteria of peptide concentration and change in critical temperature.
- “once the protein mobility within the condensate becomes fully arrested”. How do the authors measure this?
- “ According to our simulation setup, and based on the obtained trends, the process follows a nucleation-elongation mechanism [127].” I found this sentence very confusing
- It is difficult to see a reduction in the LARKS areas in Figures 4a-b. I wonder if you could add the same boxes as in Figure 4c?
- “This suggests that, even for a sequence with only two negatively charged residues and no positively charged amino acids, as FUS LCD, a potential strategy to preserve liquid-like behaviour is to insert charged molecules of the same sign as the protein that establish frequent interactions throughout the sequence (e.g., as π - π contacts) while enhancing electrostatic repulsion partially disrupting inter-protein connectivity.” This is a monstrously long sentence that I found very confusing

Reviewer #2

(Remarks to the Author)

This study presents a well-designed and innovative computational framework for identifying small peptides that regulate ageing kinetics in biomolecular condensates. By integrating equilibrium and non-equilibrium simulations, the authors provide a rigorous and systematic analysis of how peptide insertion affects the phase behaviour and ageing dynamics of TDP-43 and FUS condensates. Peptides with a precise balance of aromatic and charged residues can significantly slow ageing, offering valuable mechanistic insights into density reduction and electrostatic interactions that disrupt fibril formation. The study advances our understanding of condensate ageing and also establishes a powerful computational pipeline for rapidly screening potential therapeutic interventions. This work is highly impactful, with strong potential for further applications in disease-related phase transition studies. I only have a few minor comments.

1. Simulation of mechanical properties of condensates is very interesting. It can be more convincing and complete if the authors can show the change of G' and G'' at more time points during aging due to the formation of beta-sheets network. It will also be interesting to see how G' and G'' change with the addition of peptides.
2. Figure 2 can be improved. Fig. 2a can be modified to show the information clearer. Fig. 2b and Fig. 2c are difficult to read with many error bars. The texts in the legend are very small.
3. It might help to guide the audience by providing a table to summarize all the peptides used in the paper (e.g. length, charges, number of pie rings)
4. It is shown that certain peptides can delay the nucleation lag phase of the amyloid fibrils formation. It can be very insightful to demonstrate how peptides with different concentration can alter the lag phase. Is it possible to completely prevent the fibrils formation?
5. The writing can be shortened and simplified for broader audience.

Version 1:

Reviewer comments:

Reviewer #1

(Remarks to the Author)

I thank the reviewers for their clarifications. They are very helpful. However, the reviewers did not address some of my issues, and I believe there is still significant room for improvement in terms of readability (see below). Otherwise, I think the paper is a great piece of work that makes a substantial contribution to the advancement of the subject.

- 1) The authors find some changes in their average $t_{1/2}$, but I am not sure how significant they are due to the Poisson distribution they follow. How much do the peptides reduce the probability of presenting a rapid aging transition? On page 9, it is stated that “apart from the ageing half-time, the nucleation lag time (defined as the time at which the first inter-protein β -sheet is formed), and $-\log(kn)$ (where kn is the primary nucleation rate constant [125]) can also be used to quantify the extent to which the ageing rate is altered by the presence of peptides.” However, they do not provide us any quantitative outcome from these analyses that tells us how much the peptides are reducing the probability for an early aging transition. This information is especially relevant because in the conclusions the authors state that some of the peptides “drastically reduce the likelihood of LARKS high-density fluctuations.” Can authors give a quantification of this?
- 2) From your response: “We present only 3 representative trajectories (the closest in terms of half-time to the average half-time).” This should be clarified in the caption of Figure 3.
- 3) I think the authors need to give an intuition to the reader of what G' and G'' account for. These might be standard concepts in physics, but I guess the author’s intention is that this work will be read by scientists with a non-physics background (like myself). The same goes for the frequency (which could be clarified in the caption).
- 4) Page 5: “The chosen residues are serine (S), tyrosine (Y), and glycine (G) which ensure substantial binding affinity to the

aromatic-rich regions in FUS LCD and TDP-43 LCD sequences due to π - π contacts mediated by tyrosine." From this sentence it is not clear whether S and G behave as stickers or neutral aa.

5) Page 5: "In contrast, both E and aspartic acid (D) have been shown to play comparable roles when establishing LLPS intermolecular interactions." Which role are the authors referring to?

6) Page 9, 2nd paragraph: "decreasing $t_{1/2}$ " shouldn't be increasing $t_{1/2}$?

7) Page 10, 5th paragraph: "We find no correlation of slower D with greater $t_{1/2}$." It is not clear which effect the authors are trying to discard here.

Reviewer #2

(Remarks to the Author)

All my comments are addressed. Now the manuscript is in a much better shape, ready for publication.

Version 2:

Reviewer comments:

Reviewer #1

(Remarks to the Author)

The authors have resolved all my issues, and so I support the publication of their manuscript.

Revision of *Charged peptides enriched in aromatic residues decelerate condensate ageing driven by cross- β -sheet formation*

Ignacio Sanchez-Burgos, Andres R. Tejedor, Alejandro Castro,
Alejandro Feito, Rosana Collepardo-Guevara and Jorge R Espinosa

May 15, 2025

We thank the reviewers for their insightful comments and suggestions which have led us to improve the quality of our manuscript. We have revised the paper to address all the comments from the reviewers and respond point by point in the following pages. All changes in the manuscript have been highlighted in red.

Comments by Reviewer #1

This research presents a computational workflow to assess the impact of short peptide insertion in biomolecular condensates to mitigate their aging process (transitioning from a liquid to a solid-like state). This mechanism is associated with multiple neurodegenerative diseases, making the subject addressed here particularly important. Furthermore, the simulations elucidate the molecular principles governing the activity of these peptides, which can inspire novel designs and experiments in the future.

However, I have some concerns in relation to the time delay calculations of the ageing process. As the data is shown now, I don't think it supports the statement in the abstract: 'small peptides (...) decelerate ageing up to two orders of magnitude'. The authors find some changes in their averages, but I am not sure how significant they are due to Poisson distribution they follow. How much do the peptides reduce the probability of presenting a rapid aging transition? Figure 3 a and c show clear trends, but it is not clear which data is presented there (one selected curve or averaging all curves?). In addition, the authors say they run 6 replicas per system, but I only find 5 in some systems (e.g. Figure S1e).

Response: We thank the reviewer for the time invested in carefully assessing our manuscript and providing useful feedback. In what follows, we address the questions and concerns point by point.

Regarding the abstract statement "decelerate ageing up to two orders of magnitude" we make reference to the fact that in Figure 3d, the fastest condensate exhibiting ageing in TDP-43 LCD displays a half-time of $t_{1/2} = 119$ ns, while for the slowest in presence of (SYGSYEGS)₃, $t_{1/2} = 5500$ ns, which is over an order of magnitude (i.e., approximately 50 times slower). Moreover, we have also identified few cases in which across the whole simulation timescale (>5000 ns) no cross- β -sheet formation was observed. However, we agree with the fact that in average, the largest deceleration we observe should be estimated to be over one order of magnitude only, therefore we have amended this statement in the abstract.

Moreover, regarding Figures 3a and 3c, we present only 3 representative trajectories (the closest in terms of half-time to the average half-time of all the performed trajectories since averaging the whole curve for several trajectories produces noisy curves which may not be representative of the condensate cross- β -sheet formation) for different types of systems: (1) the protein only, (2) in presence of peptides that display moderate deceleration, such as (SYGESYGSYGEG)₂ and (SYGSYGSYEGEG)₂ for FUS LCD and (SYGSYEGS)₃ and (SYGESYGSYGEG)₂ for TDP-43 LCD, and (3) the peptide driving the strongest deceleration, as (SYGSYEGS)₃ for FUS LCD and (SYGKSYGSYGKG)₂ for TDP-43 LCD. We display the information in this manner because we have performed the total amount of 36 trajectories for each protein sequence, therefore showing all of them in the main text would result inexpedient for the reader. We believe that showing some relevant examples

in Figures 3a and 3c (i.e., the closest to the average half-time for each system), in addition to displaying a quantitative analysis of the 2x36 simulations in panels 3b and 3d is the most effective way to present a complete picture of our non-equilibrium ageing simulations. In that sense, the statistical analysis of each system is presented in Fig. 3b and 3d for FUS LCD and TDP-43 LCD, respectively, indicating the average half-time for the cross- β -sheet formation, and the 25 and 75 percentile values from our dataset of each condensate type.

Lastly, we now indicate in the Supplementary Material that for 2 of the 72 trajectories in total, we did not observe ageing, due to the computational limitation that running these simulations for extremely long timescales involves. These systems were TDP-43 LCD + (SYGESYGSYGEG)₂, and TDP-43 LCD + (SYGSYEGS)₃ in which after 5000 ns no cross- β -sheet formation was observed. For these particular cases, we have estimated the nucleation time, the aggregation half-time, and the primary nucleation rate (k_n) through the Poisson distribution formula. This is discussed in the SM Section SVI (second paragraph).

Changes: Corrected abstract.

In addition, could the authors support their findings with some experimental data? I understand that direct comparison is very difficult, but maybe there are some experiments available in the literature that produce indirect support to their findings? How do experimental G' and G'' modules compare with their simulations?

Response: We thank the reviewer for suggesting to introduce a direct comparison with experimental data relevant to our simulations. However, as foreseen by the reviewer, finding direct validation of our data is extremely difficult. Nevertheless, we have introduced an extended discussion on how the CALVADOS2 model successfully reproduces the trend of condensate viscosities upon specific sequence mutations in the low-complexity domain of another RNA-binding protein (hnRNPA1) which also exhibits ageing as FUS LCD and TDP-43 LCD (<https://doi.org/10.1371/journal.pcbi.1012737>). This trend is similarly reproduced by another model, the Mpipi-Recharged, which has also been shown by us to reproduce extremely well the drift in viscoelastic properties, and the relative viscosity, of both FUS LCD and the full-sequence of pure FUS condensates over time (<https://doi.org/10.1101/2025.03.26.645197>). Moreover, this model has also reproduced the phase-diagram and single-molecule radius of gyration of TDP-43 (<https://doi.org/10.1101/2025.02.21.639421>). Since in Ref. (<https://doi.org/10.1371/journal.pcbi.1012737>) we have shown that both models perform similarly well in predicting condensate viscosities and phase diagrams for up to 15 hnRNPA1 low-complexity domain mutants, this fact constitutes an indirect evidence that our simulations presented here provide realistic qualitative predictions of G' and G''. Finally, we note that the lack of explicit solvent and the coarse grained nature of our force field do not permit a quantitative comparison our viscosity values with experimental measurements, nevertheless, as shown in Ref. (<https://doi.org/10.1371/journal.pcbi.1012737>), both CALVADOS2 and Mpipi-Recharged accurately reproduce variations in condensate viscosity upon sequence mutations and ageing. Moreover, our results for pure FUS LCD are qualitatively consistent with the experimental in vitro results of condensate ageing found in Refs. 52 and 54. This discussion has been included in page 4 (first paragraph) of the main text.

Changes: Extended discussion in page 4 (first paragraph) of the main text.

My other criticism is that I found the presentation of the methodology very confusing and cryptic at some points, so it is very difficult for me to evaluate. This is unfortunate because the authors consider their methodology as one of the assets of their manuscript. If I tried to follow up their pipeline, I would have a myriad of questions. In the SM should be clearly explained how the different type of simulations is set up, in which conditions are performed and how many replicas (i.e. force-field, solvent model, periodic boundary conditions, number of molecules, density and/or concentration etc..). I would also find necessary to share all their scripts so that the community could really benefit from their computational pipeline.

Response: We appreciate the reviewer's concerns regarding the clarity of the methodology presentation. All the necessary information to replicate the simulations is already provided in the SM. We have ensured that all essential details are included, and additional clarifications were added where needed to improve transparency. The conditions for the simulations, including force field, and other relevant parameters, are now clearly outlined (in SM: pages 1 and for the simulation details, including force field, periodic boundary conditions, thermostat, barostat, integration

timestep, pages 2 and 3 for the ageing algorithm, and table S1 in page 5 for the concentration details).

Moreover, we have uploaded the dynamic algorithm for performing the non-equilibrium ageing simulations, configuration files and LAMMPS scripts for both FUS LCD and TDP-43 LCD condensate mixtures to ensure that other groups can easily implement our computational pipeline in a straightforward manner. This can be found following the link <https://doi.org/10.5281/zenodo.15168499>. We have linked this additional material to the main text and SM in the revised version. Any additional files can be available upon reasonable request to the corresponding authors.

Changes: Added information regarding the simulation details in the SM (pages 1, 2, 3 and 5) and included link to repository.

Another minor criticism is that I found the writing convoluted and repetitive. Each result section starts with a kind of description of the whole paper, making the text unnecessarily long. I found section IIA particularly confusing as it aims to provide a general overview but is quickly plagued with details just superficially explained. The PMF from all-atom simulations is an illustration of this point: it appears several times in the main text, but it is not till the nth occasion that it is well explained, the first mentions being totally puzzling.

Response: We would like to thank the reviewer for this helpful comment to make the manuscript clearer. We have worked in improving the main text, specially Section IIA. This point was also raised by the second reviewer, and we have addressed multiple changes which we hope help the readers to follow the manuscript. Regarding the PMF calculations, we agree that there was an excess of details on this matter, and that was not central to this manuscript since these calculations were performed previously to support the parameters used in the dynamic algorithm for modelling cross- β -sheet formation. We have modified the text in order to reflect the reviewer's suggestion, and avoid mentioning such calculations without giving more details, and doing so only when needed.

Changes: Text changed accordingly in Section IIA (page 2) and IIB (page 4).

I would also appreciate a more detailed explanation of how the authors decided the different patterns for peptide sequences.

Response: We thank the reviewer for the relevant comment about how the design of the specific peptide sequences was chosen. As shown in Figs. 2b and 2c, we have tested over 15 different sequences, all with the same length (24 residues) but with different composition and patterning. Most of them are 3-repeats of 8 residues, or 2-repeats of 12 residues. As discussed in the main text, the composition has been modulated to promote specific binding into the protein LARKS, and also for displaying moderate electrostatic self-repulsion and partially reduce the local protein LARKS high-density fluctuations driving cross- β -sheet formation. In terms of patterning, we have tested two sequences in which the same composition was maintained but their patterning modified: (SYGESYGSYGEG)₂, and (SYGSYGSYEGEG)₂. According to our results (Figs. 3b and 3d), the patterning plays a significant role in decelerating condensate ageing beyond the chosen sequence composition. However, that seems to be also dependent on the protein sequence forming the condensates, since the role of patterning for these two specific peptides is moderately different in FUS than in TDP-43, although both display the same overall trend. We have tried in all cases to select sequence patterns in which both charged and aromatic residues are partially grouped to create patches which either aim to specifically bind to the LARKS via π - π stacking or promote electrostatic self-repulsion through the accumulation of few charges close to each other. Hence, despite our work presented here should be taken as a proof-of-concept of the computational approach rather than as an extensive analysis of the optimal sequence to decelerate ageing in both protein condensates, we reveal that a combination of charged and aromatic residues seems promising to avoid condensate ageing while marginally altering their stability. An discussion on these different points are included in page 9 of the main text.

Changes: Extended relevant discussions have been included in page 9 (third paragraph).

Finally, the authors claim that their methodology is 'rapid', although they do not offer any quantification. How long do their simulations take?

Response: We agree with the reviewer that stating the simulation time does not implicitly report the actual simulated time. In our case, we are capable of performing simulations at a speed of 100 timesteps per CPU second, which allows us to simulate approximately 100 ns per day for systems of ~ 15000 particles (i.e., ~ 100 protein replicas) and 32 CPUs per simulation. We believe that reporting these specific numbers in the main text might not be relevant to all readers and that may be ambiguous for other groups that perform computer simulations using another architectures (GPUs instead or CPUs), a different amount or model of CPUs/GPUs, or another MD software. Nevertheless, since despite not being an absolute number, we agree with the reviewer that providing an estimate can be very useful for potential users. In page 3 of the Supporting Material we provide estimates for accomplishing the different simulations detailed in the manuscript, as well as a reference to this section in the main text for readers who might be interested in this performance benchmark.

Changes: We have included in page 3 of the SM specific details on the performance benchmarks using our HPC CPU server architecture for the different types of simulations performed in this study. A reference to this sections has been also provided in the main text (page 2).

List of detailed comments:

- ‘Local high-density fluctuations’: This expression appears in several places, but it is not clear to me what it means.

Response: We apologise for having included an expression which might not be clear. By local high-density fluctuations, we refer to phase-space density fluctuations in which a high concentration of multiple LARKS are found together in a small volume. This local high-density fluctuations of LARKS have been shown to promote disorder-to-order LARKS structural transitions and the subsequent formation of cross- β -sheets as discussed in the manuscript.

Changes: This concept has been further explained the first time that it is mentioned, in section IIA (first paragraph).

- NpT is not defined the first time it appears in the text. Why the authors think it is worth specifying that these simulations are N, P, T constant. Aren’t they all? Why is ‘p’ not a capital letter?

Response: We thank the reviewer for noticing these two issues. We now define NPT the first time it appears. Moreover, we have changed the notation to uppercase P. Although multiple well-established works make use of lowercase for the pressure, we agree that the uppercase use is dominant in modern day literature.

Regarding the use of constant pressure in simulations, it is not necessarily the case. Depending on the specific purpose of a MD simulation, different thermodynamic ensembles can be used, including the isothermal-isobaric ensemble (NPT), the canonical ensemble (NVT) which we use to study temperature dependent processes at constant density and compute viscoelastic properties, and others such as the microcanonical ensemble (NVE), and the grand canonical ensemble (μ VE) (which allows the exchange of particles). In our case the NPT simulations help us establish the phase boundaries, and the ageing simulations (in the canonical ensemble) allow us to study condensate maturation isolating the effects of volume fluctuations by simulating at constant density (the condensate equilibrium density). The fact that the ageing simulations are performed in the canonical ensemble is now explicitly stated in the revised version. Furthermore, for computing the viscoelastic behaviour (i.e., G' and G''), the NVT ensemble must be used, otherwise the stress tensor cannot be computed. Nevertheless, when the NVT ensemble is used, we fix the simulation density to the condensate equilibrium density in bulk conditions.

Changes: We have replaced NpT by NPT across the manuscript, and explicitly stated the thermodynamic ensemble for the ageing simulations, both in the main text (Section IIA) and SM Section SII.

- If I understood well, the NpT simulations go first in the computational pipeline, but then they are described after the ‘aging simulations’ in SM. Why? I found this really confusing.

Response: We agree with the reviewer that this might be counterintuitive, and hence, we have swapped the order of those two sections in the SM.

Changes: SII and SIII order swapped.

-How are G' and G'' calculated, and why are they obtained as a function of frequency?
Frequency of what?

Response: We apologize for not having explicitly stated more details on how these quantities are computed, and what the frequency means. We have added a new section to the SM (SIV), in which this calculation is further explained in detail. In brief, the stress relaxation modulus $G(t)$ must be first obtained from the different components of the pressure tensor and applying the Green-Kubo relation. Once obtained, G' and G'' are directly calculated from the complex modulus $G^*(\omega)$ defined as $G^*(\omega) = G'(\omega) + iG''(\omega)$. The complex modulus is calculated as the Laplace transform of $G(t)$, and thus, the storage and loss moduli can be computed as the sine and cosine transforms of $G(t)$. Please see section SIV for a detailed explanation, including the equations necessary for this calculation.

In this context, the frequency (ω) refers to the angular frequency of an oscillatory deformation that the system would experience in a rheological experiment. In experimental rheology, an oscillatory strain is applied, and the material's response is measured. This approach allows MD simulations to predict rheological behavior without explicitly running large-scale oscillatory deformation simulations. We hope that this clarifies this important aspect of the manuscript.

Changes: Section SIV added to SM.

- What is the difference between DC and NpT simulations?

Response: In Direct Coexistence simulations, both the condensed and diluted phases coexist in the same simulation box. This is achieved by placing the different biomolecules in an elongated simulation box and performing an MD simulation in the canonical (NVT) ensemble. When equilibrium is reached, both phases coexist at their corresponding equilibrium densities for the temperature fixed. In contrast, in the NPT only one phase (the condensed phase in bulk conditions) can be simulated. We fix the temperature at $P=0$ because the vapor pressure is negligible, and therefore this approach yields correct results to predict the equilibrium conditions and phase boundaries of the system from the condensed phase in mixtures including different components (please see our previous work using this method (<https://doi.org/10.1016/j.bpj.2023.03.006>) for further details). This approach only becomes a worse approximation in the vicinity of the critical temperature, as discussed in the manuscript. We opt for the NPT approach to estimate the critical temperature in presence of small peptides because such method allows a better control of the condensate composition as shown in Ref. <https://doi.org/10.1016/j.bpj.2023.03.006>.

Changes: We have added extra information to Section IIB (first paragraph) that highlights the contrast between DC and bulk NPT simulations.

- It is not clear how the critical temperature is calculated, whether using DC or NpT simulations

Response: We first determine the phase diagram of the system with the protein only (FUS or TDP-43) by means of DC simulations in combination with the laws of rectilinear diameters and critical exponents. We first opt for this method because it provides good accuracy while being computationally very efficient. Next, when we address systems in which small peptides are present, we use NPT simulations of the condensate in bulk conditions, and estimate the critical temperature as the average between the highest temperature at which the condensate is stable at $P=0$ atm and the lowest one at which the condensed phase evolves into the diluted phase. We use this criteria because in the vicinity of the critical temperature, the vapor pressure acting on the condensate will be non-negligible and the system will spontaneously evolve into the vapor phase when the critical temperature is approached. The NPT method, as explained in the previous point, enables much better control of the condensate composition as shown in Ref. <https://doi.org/10.1016/j.bpj.2023.03.006> than DC simulation in mixtures, and thus we use it for condensate mixtures. In any case, the critical temperatures predicted by both methods for the pure system agree within the uncertainty of each method ($\sim 5K$). This information is now reflected in the first paragraph of section IIB of the main text.

-‘Through the combination of DC simulations and the laws of rectilinear diameters and critical exponents [103], we determine the coexistence densities and critical solution temperature for FUS LCD pure condensates (Fig. 2(a); black filled and empty symbols, respectively).’ This should be explained in the SM.

Response: We appreciate the reviewer’s suggestion to move this explanation to the Supplementary Material. However, we believe that keeping this information in the main text is important to ensure clarity and accessibility for all readers, including those who may not be highly familiar with the methods associated to MD simulations. This ensures that the results and their interpretation are fully comprehensible to a broad audience. We hope the reviewer understands our rationale for this decision. Nevertheless, as suggested by the reviewer, we have also included a new Section SV in the SM where we further detail these calculations.

-How the authors establish which are the good criteria of peptide concentration and change in critical temperature.

Response: Our approach is characterized by the search of peptides capable of slightly decreasing the condensate critical solution temperature below $T/T_c \sim 0.05$, which typically corresponds to 5-10 K, with a consequential decrease of the condensate density in 5 % approximately. This should hinder the formation probability of LARKS high-density fluctuations. At the same time, it is also crucial to preserve the material properties of the condensates and the overall condensate composition, which in turn impacts other physicochemical parameters such as the surface tension. This is the reason why we also restrict our search to small peptide concentrations lower than 0.05 peptide/protein mass ratios. This rationale was included in Section IIB initially, however, to make it clearer, we have modified Section IIB accordingly to emphasize this concept.

Changes: Section IIB has been rewritten accordingly.

-‘once the protein mobility within the condensate becomes fully arrested’. How do the authors measure this?

Response: We thank the reviewer for this insightful comment. The direct method for ensuring that there is an effective loss of the liquid-like properties of the condensate is to measure G' and G'' upon ageing. We have done this calculation in S11 as well as in the past for multiple cases (including FUS) including references <https://doi.org/10.1002/advs.202207742> <https://doi.org/10.1038/s41467-022-32874-0> <https://doi.org/10.1101/2024.11.15.623768>. In all these works, we have analyzed how the accumulation of cross- β -sheets directly relates with protein kinetic arrest within the condensates. Where G' becomes comparable or higher than G'' at low frequencies (i.e., 10^8 rad/s), the elastic modules dominates vs. the loss one, and hence solid-like behaviour is displayed. Hence, we have modified the first paragraph of page 4 to make clearer this statement and relate the formation of cross- β -sheets to the formation of percolated cross- β -sheet networks across the condensate which preclude protein diffusion.

Changes: Sentences modified in page 4 (first paragraph) and page 8 (first paragraph) of the manuscript.

-‘According to our simulation setup, and based on the obtained trends, the process follows a nucleation-elongation mechanism [127].’ I found this sentence very confusing

Response: As detailed in the 2014 paper by Arosio *et al.* (<http://dx.doi.org/10.1016/j.tips.2013.12.005>) the nucleation-elongation mechanism implies that the formation of the first aggregate is the limiting step in the aggregation process, and once the first seed is formed, the aggregation cascade rapidly takes place. This mechanism occurs in our aggregation simulations, thereby we draw this parallelism. We agree that this was not fully explained in the manuscript, and we have amended this sentence for better clarity.

Changes: Amended sentence in page 8 of the main text.

- It is difficult to see a reduction in the LARKS areas in Figures 4a-b. I wonder if you could add the same boxes as in Figure 4c?

Response: We thank the reviewer for this helpful suggestion, which we have now implemented in the revised manuscript.

Changes: Updated Figure 4.

- ‘This suggests that, even for a sequence with only two negatively charged residues and no positively charged amino acids, as FUS LCD, a potential strategy to preserve liquid-like behaviour is to insert charged molecules of the same sign as the protein that establish frequent interactions throughout the sequence (e.g., as - contacts) while enhancing electrostatic repulsion partially disrupting inter-protein connectivity.’ This is a monstrously long sentence that I found very confusing

Response: We agree with the reviewer, and we have accordingly rephrased and split this sentence to improve its readability.

Changes: Amended sentence in page 11 of the main text.

Comments by Reviewer #2

This study presents a well-designed and innovative computational framework for identifying small peptides that regulate ageing kinetics in biomolecular condensates. By integrating equilibrium and non-equilibrium simulations, the authors provide a rigorous and systematic analysis of how peptide insertion affects the phase behaviour and ageing dynamics of TDP-43 and FUS condensates. Peptides with a precise balance of aromatic and charged residues can significantly slow ageing, offering valuable mechanistic insights into density reduction and electrostatic interactions that disrupt fibril formation. The study advances our understanding of condensate ageing and also establishes a powerful computational pipeline for rapidly screening potential therapeutic interventions. This work is highly impactful, with strong potential for further applications in disease-related phase transition studies. I only have a few minor comments.

1. Simulation of mechanical properties of condensates is very interesting. It can be more convincing and complete if the authors can show the change of G' and G'' at more time points during aging due to the formation of the beta-sheet network. It will also be interesting to see how G' and G'' change with the addition of peptides.

Response: We thank the reviewer for the extremely positive feedback and thoughtful comments on our manuscript. We completely agree that it will be illustrative to add more details on G' and G'' calculations and show results at intermediate points. Since the focus of our work in this case is less associated with a direct measurement of viscoelastic properties with ageing from simulations (which we have addressed in more detailed in our recent works: <https://doi.org/10.1038/s41467-022-32874-0> , <https://doi.org/10.1101/2025.03.26.645197> , <https://doi.org/10.1101/2025.03.11.642656> , <https://doi.org/10.1021/acs.jpcc.3c01292>) and more focused on the kinetics of the ageing process, we decide to place this new additional figure (S1) to the SM. Furthermore, we have now added an additional Section to the Supplementary Material (SIV) in which we expand on this information, as well as briefly extended this discussion in the main text. Moreover, aside from further technical details, we also discuss in the main text how a gradual transformation of these two quantities takes place as inter-protein β -sheets are established between different protein replicas. We illustrate in Figure S1 this transformation to a solid-like state with the addition of G' and G'' calculations at intermediate points. We thank the reviewer for this really good suggestion.

Moreover, as requested by the reviewer, we have measured the viscosity of a FUS LCD in presence and absence of the (SYGSYEGS)₃ peptide. This is now shown in Figure S2, where we observe how the presence of this peptide marginally affects the viscoelastic properties of the condensate, corroborating that the addition of small peptides only in moderate concentrations does not alter this crucial property of biomolecular condensates. We also make reference to this new calculation in the main text (PAGE XXXXXXXX).

Changes: Brief discussion added in page 4 (first paragraph), as well as an additional Figure S1 and calculation details in Section IV of the SM. Figure S2 added, calculating viscosity in absence and presence of small peptides.

2. Figure 2 can be improved. Fig. 2a can be modified to show the information clearer. Fig. 2b and Fig. 2c are difficult to read with many error bars. The texts in the legend are very small.

Response: We thank the reviewer for the helpful suggestions. We have increased the line width for the dashed lines in panels b and c in order to display the different trends clearer. Moreover, we have also increased the font size of the figure legend, which we agree was too small. We thank the reviewer for this useful suggestion.

Changes: Updated Figure 2.

3. It might help to guide the audience by providing a table to summarize all the peptides used in the paper (e.g. length, charges, number of pie rings)

Response: We appreciate this helpful comment, and have performed this implementation to our manuscript.

Changes: Information added to Table S2 in page 6 of the SM.

4. It is shown that certain peptides can delay the nucleation lag phase of the amyloid fibrils formation. It can be very insightful to demonstrate how peptides with different concentration can alter the lag phase. Is it possible to complete prevent the fibrils formation?

Response: We would like to thank the reviewer for this very insightful question. We have not been able to observe complete ageing inhibition mediated by the presence of small peptides, at least up to the tested peptide concentrations in which condensate stability is only moderately affected. However we have examined how varying the peptide concentration regulate the ageing process. We have included a discussion on this point at the end of the first paragraph in page 9. Moreover, we have added Section SXIII in the SM to expand on this question. We performed ageing simulations with FUS LCD in presence of a peptide that in principle leads to moderate ageing deceleration: (SYSYKRKS)₃. After doubling its concentration in the condensate, we observe a rather impactful ageing deceleration. We expect that increasing concentrations of the peptides will lead to stronger ageing deceleration and even inhibition, at the expense of altering the condensate composition and material properties.

Changes: Added discussion in the main text (page 9; first paragraph) as well as Section SXIII in the SM, addressing the reviewer comment.

5. The writing can be shortened and simplified for broader audience.

Response: We have made multiple corrections according to both of the reviewer's comments which have addressed several key points to improve the clarity and quality of the manuscript. Moreover, we have also made additional modifications to improve the readability of the manuscript, which are highlighted in red in the revised version. We hope that this revised version meets better clarity standards for a broader audience.

Revision of *Charged peptides enriched in aromatic residues decelerate condensate ageing driven by cross- β -sheet formation*

Ignacio Sanchez-Burgos, Andres R. Tejedor, Alejandro Castro,
Alejandro Feito, Rosana Collepardo-Guevara and Jorge R Espinosa

June 8, 2025

We thank the reviewer for their insightful comments and suggestions which have led us to improve the quality of our manuscript. We have revised the paper to address all the comments from the reviewer and respond point by point in the following pages. All changes in the manuscript have been highlighted in red.

Comments by Reviewer #1

I thank the reviewers for their clarifications. They are very helpful. However, the reviewers did not address some of my issues, and I believe there is still significant room for improvement in terms of readability (see below). Otherwise, I think the paper is a great piece of work that makes a substantial contribution to the advancement of the subject.

1) The authors find some changes in their average $t_{1/2}$, but I am not sure how significant they are due to the Poisson distribution they follow. How much do the peptides reduce the probability of presenting a rapid aging transition? On page 9, it is stated that "apart from the ageing half-time, the nucleation lag time (defined as the time at which the first inter-protein β -sheet is formed), and $-\log(kn)$ (where kn is the primary nucleation rate constant [125]) can also be used to quantify the extent to which the ageing rate is altered by the presence of peptides." However, they do not provide us any quantitative outcome from these analyses that tells us how much the peptides are reducing the probability for an early aging transition. This information is especially relevant because in the conclusions the authors state that some of the peptides "drastically reduce the likelihood of LARKS high-density fluctuations." Can authors give a quantification of this?

Response: We thank the reviewer for the time invested in carefully assessing our revised manuscript again. In what follows, we address the questions and concerns raised by the reviewer point by point.

Regarding the quantification of the cross- β -sheet kinetics deceleration, we have performed a Mann–Whitney U test analysis to determine whether the addition of small peptides provokes a statistically significant increase in the half-times reported in Figure 3. Through this analysis (explained in Section SVIII of the SM), we have determined that only the FUS LCD + (SYSYKRKK)₃ system is statistically alike to the pure FUS LCD condensate, while for the rest of the systems, the observed differences are statistically significant according to the analysis. We now report these values in Table S3, and discuss them in sections IIC and III. Moreover, we have modified the manuscript to avoid personal judgements such as 'drastic' in the cited example, as well as specified the ageing deceleration degree with a specific relative change, providing quantitative statistical differences with respect to the pure protein condensates.

Changes: Added and modified text in Page 9. Added Mann–Whitney U test to SM Section SVIII and Table SIII.

2) From your response: "We present only 3 representative trajectories (the closest in terms of half-time to the average half-time)." This should be clarified in the caption of Figure 3.

Response: We thank the reviewer for noticing this, we have implemented such clarification.

Changes: Added sentence to the Figure 3a caption.

3) I think the authors need to give an intuition to the reader of what G' and G'' account for. These might be standard concepts in physics, but I guess the author's intention is that this work will be read by scientists with a non-physics background (like myself). The same goes for the frequency (which could be clarified in the caption).

Response: We agree with the reviewer in this point, and have added more insight about this quantities in Page 3.

Changes: Extended explanation in Page 3.

4) Page 5: \The chosen residues are serine (S), tyrosine (Y), and glycine (G) which ensure substantial binding affinity to the aromatic-rich regions in FUS LCD and TDP-43 LCD sequences due to π -contacts mediated by tyrosine." From this sentence it is not clear whether S and G behave as stickers or neutral aa.

Response: We thank the reviewer for this insightful comment. In our design, tyrosine (Y) functions as the primary "sticker" residue due to its ability to engage in π - π interactions with other aromatic residues in the FUS LCD and TDP-43 LCD sequences, thereby mediating binding and interaction specificity. In contrast, serine (S) and glycine (G) are included as neutral or "spacer" residues. Their role is to modulate the flexibility and solubility of the peptides without significantly contributing to the intermolecular interactions driving phase separation. We have clarified this distinction in the revised text.

Changes: Revised cited sentence in page 5.

5) Page 5: \In contrast, both E and aspartic acid (D) have been shown to play comparable roles when establishing LLPS intermolecular interactions." Which role are the authors referring to?

Response: We thank the reviewer for pointing this out. To clarify, we are referring to the role of glutamic acid (E) and aspartic acid (D) as negatively charged residues that contribute to electrostatic interactions in LLPS. Specifically, these residues can mediate interactions with positively charged regions of proteins, thus contributing to the formation and stabilization of phase-separated condensates. Since the interaction parameters and size are comparable (which is not necessarily the case when comparing the different positively charged amino acids) we make this statement relevant to the discussion of the choice of the charged amino acids that we put to test. In Fig. 1 of Ref. Tejedor et al., ACS Central Science, 11, 302, 2025, we show through all-atom simulations how electrostatic interactions among glutamic or aspartic acid among themselves, or with both arginine and lysine are similar in terms of repulsive and attractive interaction strength, respectively.

Changes: We have revised the sentence to make this clearer.

6) Page 9, 2nd paragraph: \decreasing $t_{1/2}$ " shouldn't be increasing $t_{1/2}$?

Response: We thank the reviewer for noticing this mistake, we have corrected it.

Changes: Corrected sentence.

7) Page 10, 5th paragraph: \We find no correlation of slower D with greater $t_{1/2}$." It is not clear which effect the authors are trying to discard here.

Response: One of our main claims of this manuscript is how the ageing process is decelerated by the addition of peptides that generate electrostatic self-repulsion and are capable of provoking a decrease in the condensate

density and therefore, the interaction frequency between key amino acids in the ageing mechanism. Therefore, with the cited claim, we aim to discard the fact that in our condensates a variation in the protein diffusion coefficient is a crucial factor tuning ageing kinetics. We have rewritten this section accordingly to emphasize this message.

Changes: Amended sentence in page 11.